# SCALABLE IN-CONTEXT Q-LEARNING

**Jinmei Liu[1], Fuhong Liu[1], Zhenhong Sun[2], Jianye Hao[3], Huaxiong Li[1],**
**Bo Wang[1\*], Daoyi Dong[4], Chunlin Chen[1], Zhi Wang[1\*]**

[1] Nanjing University     [2] Australian National University
[3] Tianjin University     [4] University of Technology Sydney

```
jmliu@smail.nju.edu.cn    zhenhong.sun@anu.edu.au
jianye.hao@tju.edu.cn     daoyi.dong@uts.edu.au
{huaxiongli,bowangsme,clchen,zhiwang}@nju.edu.cn
```

## ABSTRACT

Recent advancements in language models have demonstrated remarkable in-context learning abilities, prompting the exploration of in-context reinforcement learning (ICRL) to extend the promise to decision domains. Due to involving more complex dynamics and temporal correlations, existing ICRL approaches may face challenges in learning from suboptimal trajectories and achieving precise in-context inference. In the paper, we propose **S**calable **I**n-**C**ontext **Q**-**L**earning (**S-ICQL**), an innovative framework that harnesses dynamic programming and world modeling to steer ICRL toward efficient reward maximization and task generalization, while retaining the scalability and stability of supervised pretraining. We design a prompt-based multi-head transformer architecture that simultaneously predicts optimal policies and in-context value functions using separate heads. We pretrain a generalized world model to capture task-relevant information, enabling the construction of a compact prompt that facilitates fast and precise in-context inference. During training, we perform iterative policy improvement by fitting a state value function to an upper-expectile of the Q-function, and distill the in-context value functions into policy extraction using advantage-weighted regression. Extensive experiments across a range of discrete and continuous environments show consistent performance gains over various types of baselines, especially when learning from suboptimal data. Our code is available at https://github.com/NJU-RL/SICQL.

## 1 INTRODUCTION

RL is a pivotal mechanism for training autonomous agents to solve complex tasks in interactive environments (Mnih et al., 2015), with expanding applications in frontier challenges such as fine-tuning language models (Yan et al., 2025; Hu et al., 2026; Zhan et al., 2026) and diffusion models (Black et al., 2024; Liu et al., 2026). A longstanding goal of RL is to learn from diverse experiences and generalize beyond its training environments, efficiently adapting to unseen situations, dynamics, or objectives (Ackley & Littman, 1989; Kirk et al., 2023; Hu et al., 2025). A promising avenue is in-context learning (Brown et al., 2020) that trains large-scale transformer models on massive datasets to achieve remarkable generalization capabilities, adapting to new tasks via prompt conditioning without any model updates (Wang et al., 2023a; Li et al., 2024a). Accordingly, in-context RL (ICRL) seeks to extend this promise to decision domains and has seen rapid progress in recent years (Nikulin et al., 2025). Existing studies contain two typical branches: algorithm distillation (AD) (Laskin et al., 2023) and decision-pretrained transformer (DPT) (Lee et al., 2023), due to their simplicity and generality. They commonly employ cross-episode transitions as few-shot prompts and train transformer-based policies under supervised pretraining (Lin et al., 2024), followed by various improvements from model-based planning (Son et al., 2025), hierarchical decomposition (Huang et al., 2024), importance weighting (Dong et al., 2025), etc (Sinii et al., 2024; Tarasov et al., 2025; Dai et al., 2024).

Though, significant challenges may emerge when extending the promise of in-context learning from (self-) supervised learning to RL, since RL involves more complex dynamics and temporal

---

\*Correspondence to Zhi Wang <zhiwang@nju.edu.cn> and Bo Wang <bowangsme@nju.edu.cn>.

correlations (Silver et al., 2021). First, previous studies usually adopt a supervised pretraining paradigm, failing to go beyond imitating collected data (Yamagata et al., 2023). AD can require long-horizon context and inherit suboptimal behaviors due to the gradual update rule (Son et al., 2025). DPT relies on an oracle for optimal action labeling that can often be infeasible in practice (Tarasov et al., 2025). These limitations may hinder the efficient learning of optimal policies, especially when only suboptimal datasets are available. Second, substantial scope remains to advance the design of efficient prompts that can precisely encode RL task information. In language communities, prompts are concise, precise text instructions rich in semantic information, as text naturally conveys high-level concepts like objects (nouns) and actions (verbs) (Kakogeorgiou et al., 2022). However, ICRL approaches generally take raw transitions as prompts that have many more tokens than a sentence and can be highly redundant. Within transitions from offline datasets, the task information can be entangled with behavior policies, thus producing biased task inference at test time (Yuan & Lu, 2022). Hence, this type of prompt may be insufficient to precisely capture relevant information about decision tasks. The aforementioned limitations raise a key question: *Can we design a scalable ICRL framework using lightweight prompts that precisely capture task-relevant information, while unleashing the core potential of fundamental reward maximization to learn from suboptimal data?*

To tackle these challenges, we draw upon two basic properties inherent in full RL. First, classical RL algorithms learn a value function to backpropagate expected returns using dynamic programming updates, [1] showing appealing stitching property, i.e., the ability to combine parts of suboptimal trajectories for finding globally optimal behaviors (Sutton et al., 1998). Naturally, the stitching property offers a compelling avenue for unleashing the potential of ICRL architectures, attaining substantial improvement over suboptimal data. Second, as RL agents learn through active interactions with the outer environment, the decision task is fully characterized by the environment dynamics, i.e., the state transition and reward functions $p(s', a|s, a)$. The world model (Hafner et al., 2025) can learn an internal representation of the environment dynamics, and is intrinsically invariant to behavior policies or collected datasets (Wang et al., 2024). Hence, leveraging the world model holds promise for designing a lightweight prompt structure capable of precisely encoding task-relevant information.

Drawing inspiration from full RL, we propose **S**calable **I**n-**C**ontext **Q**-**L**earning (**S-ICQL**), an innovative framework that harnesses dynamic programming and world modeling to steer ICRL toward efficient reward maximization and task generalization. First, we design a prompt-based multi-head transformer architecture to maintain scalability and parameter efficiency. The model simultaneously predicts optimal policies and in-context value functions using separate heads, given a task prompt and corresponding query inputs. Second, we pretrain a generalized world model to capture task-relevant information from the multi-task offline dataset, and use it to transform a small number of raw transitions into a lightweight prompt for fast and precise in-context inference. Finally, we perform iterative policy improvement by fitting a state value function to an upper-expectile of the Q-function, and distill the in-context value functions into policy extraction using advantage-weighted regression. This formulation allows for learning a policy to maximize the Q-values subject to an offline dataset constraint, while retaining the scalability and stability of the supervised pretraining paradigm. In summary, our main contributions are threefold:

- We introduce dynamic programming to supervised ICRL architectures, unleashing its potential toward learning from suboptimal trajectories with efficient reward maximization.
- We design a lightweight prompt structure that leverages world modeling to accurately capture task-relevant information, enabling fast and precise in-context inference.
- We propose a scalable and parameter-efficient ICRL framework that integrates the advantages of RL and supervised learning paradigms. Comprehensive experiments validate our superiority over a range of baselines, especially when learning from suboptimal data.

## 2  RELATED WORK

The concept of agents adapting their behaviors within the context without model updates builds on earlier work in meta-RL (Beck et al., 2025), such as the memory-based RL[2] (Duan et al., 2016)

---

[1]In this paper, we use dynamic programming to indicate the fundamental characteristic of any RL algorithm relying on the Bellman-backup operation. It updates state (or state-action) values based on value estimates of successor states (or state-action pairs), i.e., bootstrapping (Sutton et al., 1998). We use dynamic programming and Q-learning interchangeably to refer to this fundamental property of RL.

and LLIRL (Wang et al., 2022; 2023b), the optimization-based MAML (Finn et al., 2017) and MACAW (Mitchell et al., 2021), and the context-based VariBAD (Zintgraf et al., 2021), Meta-DT (Wang et al., 2024), etc. (Li et al., 2024b; Zhang et al., 2025). Recently, there has been a shift towards employing transformers to implement ICRL (Moeini et al., 2025; Nikulin et al., 2025), driven by their proven ability to capture long-term dependencies and exhibit emergent in-context learning behaviors Brown et al. (2020). AMAGO (Grigsby et al., 2024a;b) trains long-sequence transformers over entire rollouts with actor-critic learning to tackle in-context goal-conditioned problems, and (Lu et al., 2023) leverages the S4 model's ability to handle long-range sequences for ICRL tasks. In offline settings, two classical branches are AD (Laskin et al., 2023) and DPT (Lee et al., 2023). AD trains a causal transformer to autoregressively predict actions using preceding learning histories as context, while DPT predicts the optimal action based on a query state and a prompt of interaction transitions. Follow-up studies enhance in-context learning from different perspectives (Son et al., 2025; Huang et al., 2024; Dai et al., 2024; Zisman et al., 2024; Wu et al., 2025). For example, IDT (Huang et al., 2024) designs a hierarchical decision structure, and DICP (Son et al., 2025) incorporates model-based planning with a learned dynamics model. These approaches generally adopt a supervised paradigm with raw transitions as prompts, while we harness dynamic programming for reward maximization and leverage world modeling to construct more efficient prompts.

DIT (Dong et al., 2025) and IC-IQL (Tarasov et al., 2025) are the most relevant to our work. DIT also uses a weighted maximum likelihood estimation loss to train the transformer policy, where the weights are directly calculated from observed rewards in collected datasets. In contrast, we learn separate in-context value functions for computing advantage weights, enabling more stable and sample-efficient policy extraction. IC-IQL integrates a Q-learning objective into AD. The policy and value networks are updated by optimizing only their heads, without propagating gradients to the transformer backbone. In contrast, we train the full multi-head transformer policy end-to-end, releasing the transformer's scalability with a simplified pipeline. We also design a precise, lightweight prompt structure to overcome the limitation of AD algorithms that require long training histories as context. Empirical results in Sec. 4 demonstrate our superiority to these two baselines.

## 3 METHOD

In this section, we present **S-ICQL** (**S**calable **I**n-**C**ontext **Q**-**L**earning), an innovative ICRL framework that leverages dynamic programming and world modeling for efficient reward maximization and task generalization. We adopt a prompt-based multi-head transformer architecture that simultaneously predicts optimal policy and in-context value functions using separate heads, ensuring scalability and parameter efficiency. Figure 1 illustrates the method overview. The algorithm pseudocodes are presented in Appendix B, and detailed implementations are given as follows.

### 3.1 PROBLEM STATEMENT

We consider a multi-task offline RL setting, where tasks follow a distribution $M^i = \langle \mathcal{S}, \mathcal{A}, \mathcal{T}^i, \mathcal{R}^i, \gamma \rangle \sim P(M)$. Tasks share the same state-action spaces $\mathcal{S}, \mathcal{A}$ but differ in reward functions $\mathcal{R}$ or transition dynamics $\mathcal{T}$. An offline dataset $\mathcal{D}^i = \sum_j (s_j^i, a_j^i, r_j^i, s_{j'}^i)$ is collected by arbitrary behavior policies for each task out of a total of $N$ training ones. The agent can only access the offline datasets $\{\mathcal{D}^i\}_{i=1}^N$ to train an in-context policy as $\pi_\theta(a^i|s^i; \beta^i)$, where $\beta^i$ is a prompt that encodes task-relevant information (e.g., past interaction history in AD or transition sequences in DPT). During testing, the trained policy is evaluated on unseen tasks sampled from $P(M)$ through direct interaction with the environment. With fixed policy parameters $\theta$, all adaptations occur through the prompt/context $\beta$ that is initially empty and gradually constructed from history interactions. As $\beta$ evolves, the model refines its predictions, analogous to policy updates in conventional RL. The objective is to learn an in-context policy that maximizes the expected episodic return over test tasks as $J(\pi) = \mathbb{E}_{M \sim P(M)}[J_M(\pi)]$.

### 3.2 SCALABLE MODEL ARCHITECTURE

**World Modeling.** As shown in Figure 1-(a), we design a lightweight prompt structure capable of encoding precise information about decision tasks. The world model, representing environment dynamics $p(s', r|s, a)$ (Hafner et al., 2025), fully characterizes the underlying task and remains invariant to behavior policies or the datasets collected. Inspired by this fundamental property of RL,

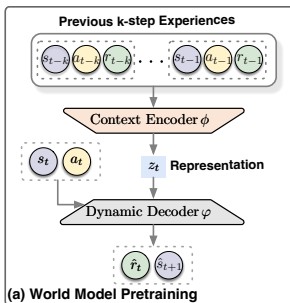 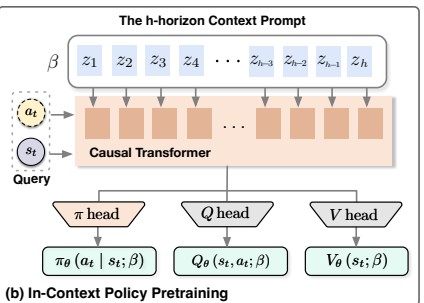 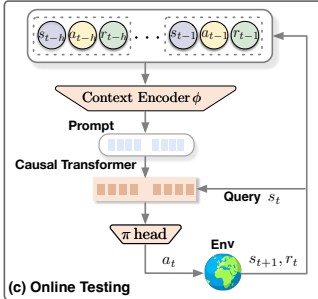

Figure 1: The overview of S-ICQL. (a) We pretrain a generalized world model to accurately capture task-relevant information from the multi-task offline dataset as in Eq. (3), and use the context encoder to transform a small number of raw transitions into a precise and lightweight prompt $\beta$ as in Eq. (1). (b) We design a prompt-based multi-head transformer model that simultaneously predicts the optimal policy $\pi_\theta(a|s;\beta)$, the state value function $V_\theta(s;\beta)$, and Q-function $Q_\theta(s,a;\beta)$ using separate heads, given the task prompt $\beta$ and corresponding query inputs ($s$ or $s, a$). We learn $V_\theta$ by expectile regression as in Eq. (5), and use it to compute Bellman backups for training $Q_\theta$ as in Eq. (4). The in-context value functions are distilled into policy extraction using advantage-weighted regression as in Eq. (6). (c) Online testing by interacting with the environment. The prompt is initially empty and gradually constructed from history interactions using the pretrained context encoder.

we pretrain a generalized world model to acquire task-relevant information from the multi-task offline dataset. The world model contains a context encoder $E_\phi$ that abstracts recent $k$-step experiences $\eta_t^i = (s_{t-k}, a_{t-k}, r_{t-k}, ..., s_{t-1}, a_{t-1}, r_{t-1}, s_t, a_t)^i$ into a task representation as $z_t^i = E_\phi(\eta_t^i)$, and a dynamics decoder $D_\varphi$ that predict the instant reward and next state conditioned on the task representation as $[\hat{r}_t, \hat{s}_{t+1}] = D_\varphi(s_t, a_t; z_t^i)$. Then, for training the in-context policy, we sample a short $h$-step trajectory randomly from dataset $\mathcal{D}^i$, and use the pretrained $E_\phi$ to transform raw transitions in this trajectory into compact task representations as

$$\beta^i := [z_1^i, z_2^i, ..., z_h^i] = \left[E_\phi(\eta_1^i), E_\phi(\eta_2^i), ..., E_\phi(\eta_h^i)\right], \quad \forall \text{ task } i. \tag{1}$$

Then, we construct a *lightweight* prompt $\beta$ using the $h$-step task representation for fast and precise in-context inference, as opposed to AD algorithms that can require long training histories as context. Sec. 3.3 presents the world model and its pretraining process in detail.

**Dynamic Programming.** Classical RL algorithms learn a value function to backpropagate expected returns using dynamic programming updates, showcasing high stitching capacity that allows for not only imitating collected data but also achieving substantial improvement beyond it (Sutton et al., 1998). Inspired by this, we harness the stitching property to offer a promising avenue for unleashing ICRL's potential toward explicit reward maximization. Following common practice in advanced offline RL algorithms (Kostrikov et al., 2022; Snell et al., 2023), we learn both a state value function $V$ and an action value function $Q$. As shown in Figure 1-(b), we design a prompt-based multi-head transformer architecture to simultaneously predict the optimal and value functions. Let $\theta$ denote parameters of the integrated transformer model. The model outputs the policy $\pi_\theta(a|s;\beta)$, the state value $V_\theta(s;\beta)$, and the action value $Q_\theta(s,a;\beta)$ using three separate heads, given the task prompt $\beta$ and corresponding query inputs ($s$ or $s, a$). Sec. 3.4 presents the detailed learning process.

This flexible design maintains two appealing properties. One is **parameter efficiency**. Our model only introduces two additional lightweight heads compared to traditional ICRL methods such as DPT, and the resulting increase in parameters is negligible relative to the foundational transformer backbone. Another is **model scalability**. Our central component remains an end-to-end causal transformer architecture that holds the promise for training RL models at scale. We learn the in-context policy by advantage-weighted regression, a supervised learning style with a simple and convergent maximum likelihood loss (refer to Eq. (6)). It ensures that our method preserves the scalability and stability inherent to the supervised pretraining paradigm. Moreover, we train the whole multi-head transformer policy end-to-end (refer to Sec. 3.5), releasing the transformer's scalability with a simplified pipeline.

### 3.3 World Model Pretraining for Prompt Construction

The environment dynamics, i.e., the reward and state transition functions $p(s', r|s, a)$, can share some common structure across the task distribution. For each task $M^i$, we approximate its dynamics by a generalized world model $P$ that is shared across tasks, defined as

$$P_i(r_t, s_{t+1}|s_t, a_t) \approx P(r_t, s_{t+1}|s_t, a_t; z_t^i), \quad \forall \text{ task } i. \tag{2}$$

As the true task identity is unknown, we infer task representation $z_t^i$ from the agent's $k$-step experience within task $M^i$ as $\eta_t^i = (s_{t-k}, a_{t-k}, r_{t-k}, ..., s_{t-1}, a_{t-1}, r_{t-1}, s_t, a_t)^i$. The intuition is that the true task belief can be inferred from the agent's history interactions, similar to recent meta-RL studies (Zintgraf et al., 2021; Ni et al., 2023). We use a context encoder $E_\phi$ to abstract recent $k$-step experiences into a task representation as $z_t^i = E_\phi(\eta_t^i)$, which is augmented into the input of a dynamics decoder $D_\varphi$ to predict the instant reward and next state as $[\hat{r}_t, \hat{s}_{t+1}] = D_\varphi(s_t, a_t; z_t^i)$. Based on the assumption that tasks with similar contexts will behave similarly (Lee et al., 2020), our generalized world model can extrapolate meta-level knowledge across tasks by precisely capturing task-relevant information. The context encoder and dynamics decoder are jointly trained by minimizing the reward and state transition prediction error as

$$\mathcal{L}(\phi, \varphi) = \mathbb{E}_{\eta_t^i \sim M^i} \left[ \|[r_t, s_{t+1}] - D_\varphi(s_t, a_t; z_t^i)\|_2^2 \mid z_t^i = E_\phi(\eta_t^i) \right], \quad \forall \text{ task } i. \tag{3}$$

After proper pretraining, we freeze the generalized world model for prompt construction in Eq. (1).

### 3.4 In-Context Q-Learning

**In-Context Value Functions.** The Q-function is trained to minimize the Bellman error as

$$\mathcal{L}_Q(\theta) = \mathbb{E}_{(s_t^i, a_t^i, s_{t+1}^i) \sim \mathcal{D}^i} \left[ \left( r(s_t^i, a_t^i) + \gamma V_\theta\left(s_{t+1}^i; \beta^i\right) - Q_\theta(s_t^i, a_t^i; \beta^i) \right)^2 \right], \quad \forall \text{ task } i. \tag{4}$$

The in-context state value function $V_\theta$ aims to fit an upper-expectile of the Q-function, and is trained to minimize an expectile regression loss as

$$\mathcal{L}_V(\theta) = \mathbb{E}_{(s_t^i, a_t^i) \sim \mathcal{D}^i} \left[ L_2^\omega \left( Q_{\hat{\theta}}(s_t^i, a_t^i; \beta^i) - V_\theta(s_t^i; \beta^i) \right) \right], \quad \forall \text{ task } i, \tag{5}$$

where $L_2^\omega(u) = |\omega - \mathbb{1}(u < 0)| \cdot u^2$ is an asymmetric loss function with a expectile parameter $\omega \in (0.5, 1)$. This form of expectile regression reduces the influence of $Q < V$ predictions by a factor of $1 - \omega$ while assigning more importance to $Q > V$ predictions by a factor of $\omega$. In this way, we predict an upper-expectile of the temporal-difference target that approximates the maximum of $r(s_t^i, a_t^i) + \gamma Q_{\hat{\theta}}\left(s_{t+1}^i, a_{t+1}^i; \beta^i\right)$ over actions $a_{t+1}^i$ constrained to the dataset actions. More details can be found in Implicit Q-Learning (Kostrikov et al., 2022).

**In-Context Policy Extraction.** The value function learning procedure allows for stitching suboptimal trajectories to discover globally optimal behaviors. Then, we distill the in-context value functions into policy extraction with advantage-weighted regression (Peng et al., 2019), a supervised learning style that uses a simple and convergent maximum likelihood loss function as

$$\mathcal{L}_\pi(\theta) = -\mathbb{E}_{(s_t^i, a_t^i) \sim \mathcal{D}^i} \left[ \exp\left( \frac{1}{\lambda} \left( Q_{\hat{\theta}}(s_t^i, a_t^i; \beta^i) - V_\theta(s_t^i; \beta^i) \right) \right) \cdot \log \pi_\theta(a_t^i \mid s_t^i; \beta^i) \right], \forall \text{ task } i, \tag{6}$$

where $\lambda > 0$ is a temperature parameter. Using weights from in-context value functions, the objective is not merely to clone behaviors from the dataset but to learn policies that maximize Q-values under a distribution constraint from dataset actions. This formulation aims to select and stitch optimal actions in the dataset while retaining the scalability and stability of the supervised pretraining paradigm.

### 3.5 Overall Optimization

To unify policy learning and value updating within a single architecture, the transformer backbone and its three specific heads are jointly optimized. This joint training objective combines supervised training with dynamic programming updates, allowing the model to simultaneously learn in-context policies and value functions while retaining scalability and stability. The overall loss is defined as

$$\mathcal{L}(\theta) = \mathsf{c}_1 \, \mathcal{L}_\pi(\theta) + \mathsf{c}_2 \, \mathcal{L}_Q(\theta) + \mathsf{c}_3 \, \mathcal{L}_V(\theta), \tag{7}$$

where we set coefficients $(\mathsf{c}_1, \mathsf{c}_2, \mathsf{c}_3)$ to a balanced ratio of $(1:1:1)$ in all experiments. A detailed analysis of how different coefficient choices affect performance is provided in Appendix H. This design ensures that S-ICQL integrates lightweight prompt construction with a unified optimization pipeline, achieving efficient reward maximization and robust generalization across tasks.

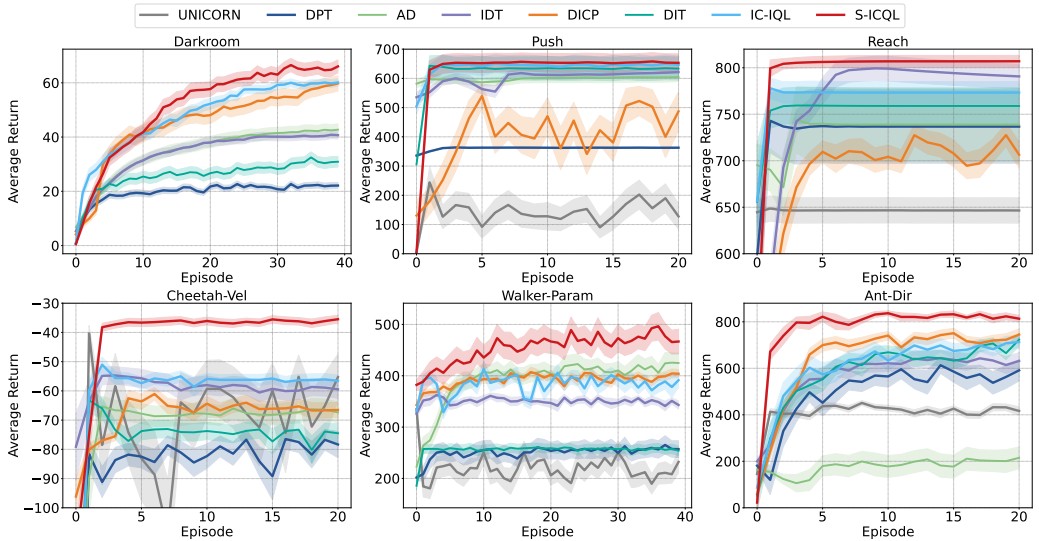

Figure 2: Few-shot evaluation return curves of S-ICQL and baselines on Mixed datasets.

# 4 EXPERIMENTS

We comprehensively evaluate the in-context learning capacity of S-ICQL on popular benchmarking domains across different dataset types. In general, we aim to answer the following questions:

- Can S-ICQL consistently outperform other strong baselines on unseen tasks? (Sec. 4.1)
- How do the prompt construction via world modeling and the reward maximization via dynamic programming affect the in-context learning performance, respectively? (Sec. 4.2)
- Can S-ICQL also achieve well performance for out-of-distribution (OOD) tasks? (Sec. 4.3)
- Does S-ICQL really have the stitching capacity to find globally optimal policy? (Sec. 4.4)
- Is S-ICQL robust to the quality of offline datasets and hyperparameters? (Sec. 4.5 and Appendix G)
- Can S-ICQL encode task-relevant information for efficient prompt construction? (Sec. 4.6)
- Does S-ICQL maintain strong performance on more complex and diverse environments? (Sec. 4.7)

**Environments.** We conduct experiments on three challenging benchmarks commonly used for evaluating ICRL algorithms: (i) DarkRoom Laskin et al. (2023), a 2D discrete environment where the agent must locate an unknown goal. (ii) MuJoCo Todorov et al. (2012), a standard testbed including tasks with varying reward functions and transition dynamics. (iii) Meta-World ML1 Yu et al. (2020), a robotic benchmark with 50 manipulation tasks. Tasks are randomly sampled and split into training sets $M^{\text{train}}$ and test sets $M^{\text{test}}$. Further environment details are provided in Appendix C.

**Pretraining Datasets.** For DarkRoom, suboptimal datasets are generated using a noisy action selection strategy by combining optimal and random policies. For MuJoCo and Meta-World, datasets are collected using a single task RL policy for each task. We construct two qualities of offline datasets: `Mixed` and `Medium`. More details on dataset construction are provided in Appendix D.

**Baselines.** We compare to six competitive ICRL approaches and one offline meta-RL method, including: 1) `IC-IQL` (Tarasov et al., 2025), 2) `DIT` (Dong et al., 2025), 3) `DICP` (Son et al., 2025), 4) `IDT` (Huang et al., 2024), 5) `AD` (Laskin et al., 2023), 6) `DPT` (Lee et al., 2023), and 7) `UNICORN` (Li et al., 2024b). Details about these baselines are provided in Appendix E.

To ensure a fair comparison, we conduct a few-shot evaluation for all methods. During testing, each policy directly interacts with the environment for a few episodes using fixed parameters, conditioned on a prompt sampled from past interactions. Results are reported as the mean of 10 trials with 95% bootstrapped confidence intervals, and standard errors are also provided.

Table 1: Few-shot evaluation returns of S-ICQL and baselines on Mixed datasets, i.e., numerical results of converged performance from Figure 2.

| Methods | Darkroom | Push | Reach | Cheetah-Vel | Walker-Param | Ant-Dir |
|---|---|---|---|---|---|---|
| UNICORN | / | 127.69± 44.04 | 646.52± 13.60 | -55.29± 7.90 | 232.23± 21.73 | 416.35± 18.29 |
| DPT | 22.12± 1.09 | 362.74± 1.91 | 736.72± 39.42 | -78.35± 4.50 | 257.11± 16.36 | 591.31± 45.54 |
| AD | 42.72± 2.14 | 604.50± 15.52 | 738.96± 39.38 | -67.37± 3.53 | 424.82± 19.23 | 215.01± 46.61 |
| IDT | 40.70± 1.44 | 621.58± 11.16 | 790.68± 13.01 | -59.46± 3.10 | 343.01± 7.95 | 631.83± 35.29 |
| DICP | 59.76± 2.80 | 487.28± 66.33 | 706.46± 14.32 | -66.53± 2.58 | 403.90± 8.09 | 745.05± 24.12 |
| DIT | 30.90± 1.96 | 633.58± 39.22 | 758.92± 19.13 | -74.50± 3.24 | 253.94± 9.04 | 723.49± 27.36 |
| IC-IQL | 60.12± 1.33 | 646.08± 30.34 | 773.33± 11.79 | -56.53± 1.98 | 391.38± 13.97 | 713.26± 27.23 |
| **S-ICQL** | **66.05**± 2.37 | **653.04**± 31.22 | **806.97**± 6.35 | **-35.48**± 1.33 | **466.72**± 24.06 | **813.34**± 14.12 |

Table 2: Offline evaluation returns of S-ICQL and baselines on Mixed datasets.

| Methods | Darkroom | Push | Reach | Cheetah-Vel | Walker-Param | Ant-Dir |
|---|---|---|---|---|---|---|
| DPT | 29.02±9.98 | 410.16±107.03 | 783.49±22.13 | -65.92±46.74 | 221.84±103.76 | 667.34±243.76 |
| DIT | 41.82±13.22 | 495.41±136.70 | 805.57±13.10 | -61.11±35.96 | 259.07±69.15 | 744.24±144.69 |
| **S-ICQL** | **76.63**±19.03 | **575.07**±118.30 | **818.04**±8.66 | **-37.87**±10.76 | **414.65**±120.22 | **835.98**±117.25 |

## 4.1 MAIN RESULTS

The baselines typically work under the few-shot setting, since they rely on task prompts or warm-start data to infer task representations. We evaluate S-ICQL against these baselines in an aligned few-shot setting, where all methods utilize the same number of interaction trajectories for task inference. Figure 2 illustrates the evaluation return curves across various environments using Mixed datasets, and Table 1 summarizes the numerical results of converged performance. Across environments with varying reward functions and transition dynamics, S-ICQL consistently demonstrates superior data efficiency and higher asymptotic performance. In more complex environments such as Ant-Dir and HalfCheetah-Vel, the advantage of S-ICQL is more pronounced, highlighting its promising in-context learning abilities on challenging tasks. Moreover, S-ICQL generally exhibits lower variance during the testing phase, indicating both improved data efficiency and better training stability. Notably, DPT yields suboptimal performance across most environments, underscoring the critical role of incorporating fundamental reward maximization into our framework. The significant improvement over DIT and IC-IQL also verifies that our method can provide a more efficient way for policy extraction with advantage-weighted regression or incorporating Q-learning objectives.

We also conduct an offline evaluation to assess the robustness of S-ICQL on trajectories collected by external behavior policies. Specifically, we use trajectories generated by non–S-ICQL policies and construct offline test sets by randomly sampling ten trajectories per task. We then evaluate DPT-style baselines on the same sampled trajectories to ensure a fair comparison. As shown in Table 2, S-ICQL maintains strong performance in the offline setting, indicating that its advantages persist even when interaction is restricted and evaluation relies solely on externally collected data.

## 4.2 ABLATION STUDY

To assess the respective contribution of each component, we compare S-ICQL with three ablations: (i) `w/o_c`, which removes the world modeling component and directly uses a trajectory of raw transitions to construct prompts; (ii) `w/o_q`, which removes the Q-learning component and learn in-context policies in a pure supervised paradigm; and (iii) `w/o_cq`, which removes both world modeling and Q-learning, reducing the model to the original DPT. In all ablations, the remaining structural components are kept identical to those in the full S-ICQL.

Table 3: Few-shot converged returns of S-ICQL and its ablations on Mixed datasets.

| Ablation | Reach | Cheetah-Vel | Ant-Dir |
|---|---|---|---|
| `w/o_cq` | 736.72 ± 39.42 | -78.35 ± 4.50 | 591.31 ± 45.54 |
| `w/o_c` | 792.09 ± 6.66 | -56.19 ± 1.94 | 693.87 ± 30.41 |
| `w/o_q` | 752.41 ± 24.15 | -63.66 ± 8.26 | 784.07 ± 24.88 |
| **S-ICQL** | **806.97** ± 6.35 | **-35.48** ± 1.33 | **813.34** ± 14.12 |

Table 4: Few-shot evaluation returns of S-ICQL and baselines for OOD tasks on Mixed datasets, i.e., numerical results of converged performance from Figure 4.

| Method | UNICORN | DPT | AD | IDT | DICP | DIT | IC-IQL | **S-ICQL** |
|--------|---------|-----|----|----|------|-----|--------|------------|
| Cheetah-Vel | -258.39± 43.67 | -137.26 ± 16.28 | -112.54± 31.92 | -103.25± 1.74 | -116.53± 2.66 | -113.18± 2.33 | -101.89± 3.50 | **-83.45**± 2.58 |
| Ant-Dir | 367.46± 15.43 | 205.29 ± 40.00 | 158.78± 5.17 | 519.80 ± 67.70 | 579.61± 51.25 | 454.29± 38.93 | 540.20 ± 47.22 | **664.95**± 28.95 |

Figure 3 shows the few-shot evaluation return curves of S-ICQL and its ablations on Mixed datasets across representative environments. Table 3 summarizes the numerical results of converged performance. First, ablating the world model causes a notable performance drop, especially in complex tasks like Ant-Dir, emphasizing its importance for precise in-context inference. Second, removing Q-learning reduces performance, underscoring its role in refining policies using suboptimal data. Integrating Q-learning enables S-ICQL to improve policies from noisy or inferior trajectories, enhancing its stitching ability to maximize rewards. Finally, removing both components results in the greatest decline, reducing the model to simple behavioral cloning that fails to generalize in complex tasks. Overall, the ablation results validate S-ICQL's effectiveness in harnessing both world modeling and Q-learning for efficient reward maximization and generalization across tasks.

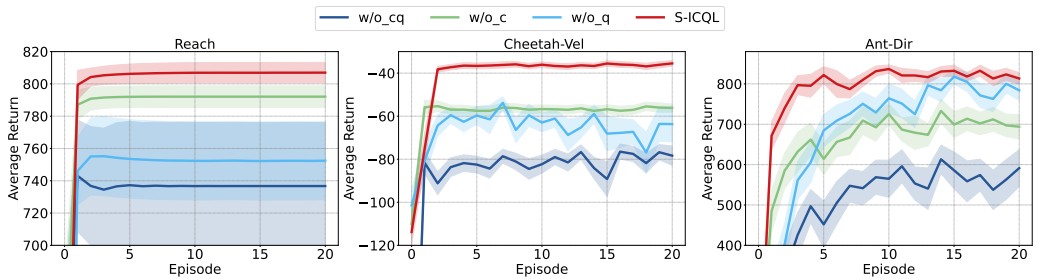

Figure 3: Few-shot evaluation return curves of S-ICQL and its ablations on Mixed datasets. `w/o_c` removes world modeling, `w/o_q` removes Q-learning, and `w/o_cq` removes both components.

### 4.3 GENERALIZATION TO OUT-OF-DISTRIBUTION (OOD) TASKS

Acknowledging the importance of evaluating ICRL algorithms under distribution shifts, we test S-ICQL against all baselines on OOD tasks across representative environments. The target velocities for Cheetah-Vel range from $[0.1, 3.0]$, and the target directions for Ant-Dir range from $[0, 2\pi]$. We split tasks by target values to construct an OOD setting. The model is trained on tasks with a lower range of target velocities/directions and tested on the remaining higher range, ensuring that training and testing tasks come from different distributions. This setup allows us to assess S-ICQL's ability to

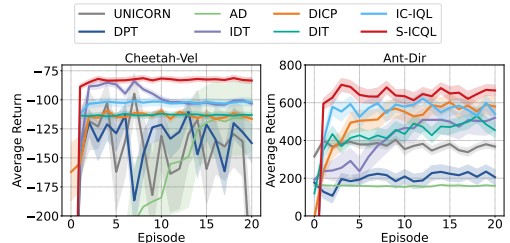

Figure 4: Few-shot evaluation curves of S-ICQL and baselines for OOD tasks on Mixed datasets.

generalize beyond the set of training tasks. As shown in Figure 4 and Table 4, S-ICQL consistently outperforms baselines on OOD tasks. We extrapolate the meta-level knowledge across tasks by the extrapolation ability of the world model, which is more accurate and robust since the world model is intrinsically invariant to behavior policies or collected datasets. The world model shares some common structure across the task distribution (even for OOD tasks), e.g., the kinematics principle or locomotion skills. Hence, the extrapolation of the world model also works for OOD test tasks in this case. This observation is also consistent with the visualization results in Sec. 4.6.

### 4.4 STITCHING CAPABILITY

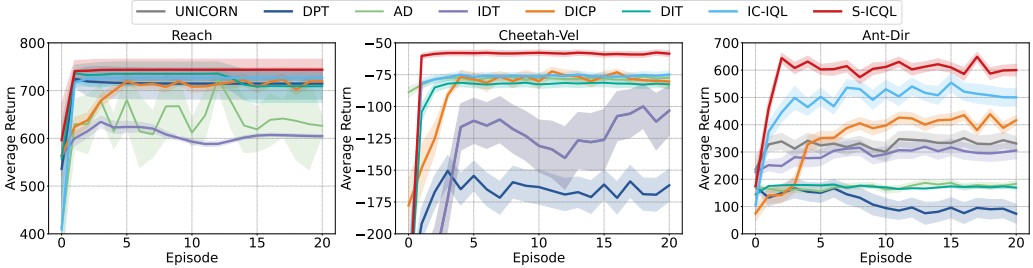

Figure 6: Few-shot evaluation return curves of S-ICQL and baselines on Medium datasets.

We conducted an in-depth analysis to examine whether S-ICQL truly exhibits stitching capability. Specifically, we pretrained the model on offline datasets containing only suboptimal trajectories and evaluated it on the corresponding training tasks. Figure 5 compares the best returns in the offline datasets (OPT) with the performance of DPT, `w/o_q` (S-ICQL without Q-learning), and S-ICQL, including both evaluation curves and final returns. As `w/o_q` and DPT follow the supervised pretraining paradigm, their performance never exceeds the dataset best, underscoring the limitations of purely supervised methods. In contrast, S-ICQL achieves performance exceeding the best in-dataset return on training tasks, providing strong evidence of genuine stitching by composing suboptimal trajectory segments into globally superior policies.

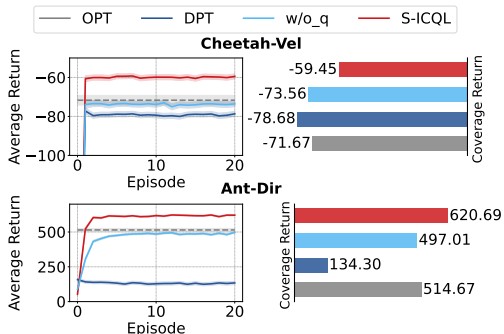

Figure 5: Comparison of best dataset returns with DPT, `w/o_q`, and S-ICQL on training tasks.

### 4.5 ROBUSTNESS TO THE QUALITY OF OFFLINE DATASETS

To evaluate S-ICQL's robustness to data quality, we conduct experiments on Medium datasets containing only suboptimal data. As shown in Figure 6 and Table 5, S-ICQL outperforms all baselines, particularly in complex environments like Cheetah-Vel and Ant-Dir, showing its superiority when learning from suboptimal data. Notably, many baselines converge to suboptimal policies using imperfect data, especially for pure imitation learning methods like DPT. This again highlights our advantage of using Q-learning to not only simply imitate collected data, but also to combine parts of suboptimal trajectories for finding globally optimal behaviors. These results, along with those in Sec. 4.1, confirm the robustness of our method to varying dataset quality.

Table 5: Few-shot converged results of S-ICQL and baselines on Medium datasets.

| Methods | Reach | Cheetah-Vel | Ant-Dir |
|---|---|---|---|
| UNICORN | 366.53± 1.98 | -103.23± 19.35 | 331.23± 25.75 |
| DPT | 714.70± 32.51 | -161.88± 12.61 | 73.72± 34.71 |
| AD | 626.24± 19.08 | -77.30± 2.06 | 183.51± 11.82 |
| IDT | 604.49± 3.54 | -103.23± 19.35 | 305.50± 30.20 |
| DICP | 719.54± 10.20 | -80.34± 5.82 | 416.58± 16.80 |
| DIT | 709.07± 33.65 | -82.61± 3.15 | 169.22± 5.18 |
| IC-IQL | 726.68± 15.61 | -74.88± 2.78 | 500.20± 31.49 |
| **S-ICQL** | **743.72**± 22.15 | **-58.48**± 2.32 | **600.18**± 21.46 |

### 4.6 VISUALIZATION INSIGHTS

We gain deep insights into the prompt construction process through t-SNE visualization on Cheetah-Vel and Ant-Dir tasks, as shown in Figure 8. We use a continuous color spectrum to indicate task similarity. For each task, we randomly sample 200 transitions and encode them into task representations using the pretrained world model. The initially entangled transitions are transformed into well-separated clusters in the prompt space, where points from different tasks are clearly distinguished and similar tasks are grouped more closely. Further, representations of Cheetah-Vel form a clear rectilinear distribution from blue (low velocity) to red (high velocity), exactly aligning

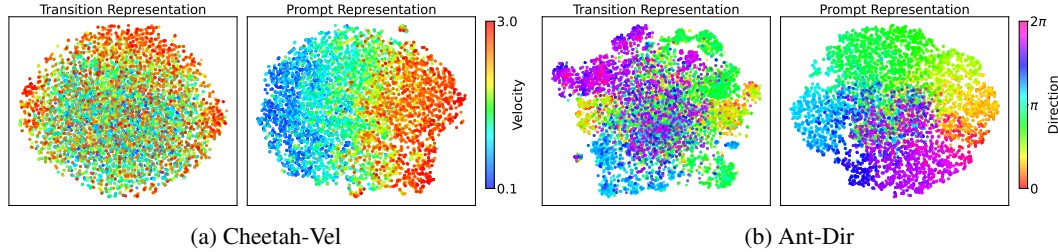

(a) Cheetah-Vel                              (b) Ant-Dir

Figure 8: t-SNE visualization on Cheetah-Vel and Ant-Dir, where tasks differ in target velocities of $[0.1, 3.0]$ and target directions of $[0, 2\pi]$. Data representations of raw transitions $(s, a, r, s')$ and precise prompts $\beta$ from a distribution of tasks are mapped into rainbow-colored points.

with the rectilinear spectrum of target velocities *in a physical sense*. Similarly, representations of Ant-Dir follow a cyclic spectrum that matches the periodicity of angular directions in a physical sense. This finding highlights S-ICQL's ability to harness world modeling to distill meaningful task-specific information from raw transitions, enabling precise task inference to facilitate in-context learning.

### 4.7 ADDITIONAL RESULTS ON MORE COMPLEX ENVIRONMENT

To further assess the scalability and robustness of S-ICQL in demanding scenarios, we extend our empirical evaluation to two challenging environments: (1) a sparse-reward, hard-exploration task (PickPlace) and (2) a high-dimensional control task (Humanoid-Dir), further increasing the diversity and difficulty of our evaluation. As shown in Figure 7 and Table 6, S-ICQL consistently outperforms all baselines, demonstrating robustness and generalization in these environments. Notably, the performance gap becomes even more pronounced in the high-dimensional setting, highlighting the particular strength of

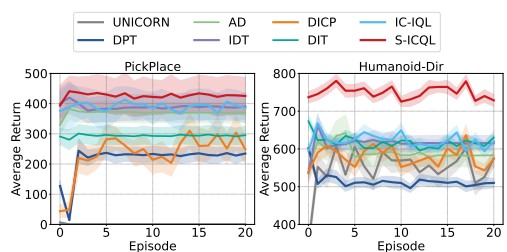

Figure 7: Few-shot evaluation return curves of all methods on Mixed datasets for complex tasks.

S-ICQL when dealing with large-scale problems, further underscoring its broader applicability.

Table 6: Few-shot converged returns of S-ICQL and baselines on Mixed datasets for complex tasks.

| Method | UNICORN | DPT | AD | IDT | DICP | DIT | IC-IQL | **S-ICQL** |
|---|---|---|---|---|---|---|---|---|
| PickPlace | 0.43± 0.00 | 234.78± 20.85 | 370.09± 33.45 | 390.62± 42.99 | 248.54± 48.20 | 295.61± 29.79 | 386.20± 47.55 | **425.30**± 61.68 |
| Humanoid-Dir | 575.00± 22.62 | 509.99± 12.40 | 583.08± 29.64 | 615.17± 20.23 | 573.48± 21.18 | 630.06± 15.39 | 620.12± 17.17 | **728.63**± 15.54 |

## 5 CONCLUSIONS, LIMITATIONS, AND FUTURE WORK

In the paper, we propose S-ICQL, an innovative framework that introduces dynamic programming and world modeling to enable fundamental reward maximization and efficient task generalization in ICRL. S-ICQL employs a multi-head transformer to jointly predict optimal policies and in-context value functions, guided by a pretrained world model that encodes precise task-relevant information for efficient prompt construction. Policy improvement is achieved by fitting in-context value functions with expectile regression and extracting policies via advantage-weighted regression, enabling reward maximization while preserving the scalability and stability of supervised pretraining. Extensive evaluations verify the consistent superiority of S-ICQL over a range of competitive baselines.

Though, our prompt length matches the sampled transitions, which may be too long for long-horizon interactive problems. Future work can explore using more compact task representations, e.g., encoding an episode or a skill into a single token. Another promising step is to leverage natural language as a higher-level task prompt for ICLR. We also plan to utilize the dynamics decoder at test time to detect distributional shifts, facilitating adaptive learning through continual updates.

## ACKNOWLEDGEMENTS

This work was supported in part by the National Natural Science Foundation of China (Nos. 62376122 and 72471109), in part by the National Key Research and Development Program of China (No. 2025YFA1016904), and in part by the National Natural Science Foundation of China (Nos. 92370132 and 72471109).

## ETHICS STATEMENT

We are not aware of any major ethical concerns arising from our work. Our study is conducted entirely within RL benchmarking environments, using only publicly available models and datasets for training and evaluation. No human subjects were involved, and our research does not introduce sensitive or potentially harmful insights.

## REPRODUCIBILITY STATEMENT

We provide detailed descriptions of our experimental settings in the Appendix, including environments, datasets, implementation details, and hyperparameters. In addition, we provide the original code in supplementary materials to facilitate reproducibility and verification of our results.

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

## A  LIMITATIONS

Despite promising results, our approach has several limitations that warrant further investigation:

**Prompt Length and Efficiency:** Although the prompt length matches the sampled transitions, it may not be compact enough for long-horizon tasks, where increased complexity can cause inefficiencies. Future work should focus on condensing the prompt without losing key context.

**Task Information Extraction:** The current approach uses prompts equal in size to the sampled transitions, which works for short tasks but may not scale well for more complex ones. Using compact representations, like encoding episodes or skills into single tokens, could improve scalability.

**Leveraging Natural Language for S-ICQL:** A promising direction is using natural language as a higher-level prompt for S-ICQL. Pretrained language models could enhance knowledge transfer, improving generalization and adaptability, offering valuable opportunities for future research.

These limitations highlight critical areas for improvement, particularly in reducing prompt size and enhancing scalability for long-horizon tasks, offering valuable avenues for future research.

## B  ALGORITHM PSEUDOCODES

Based on the implementations presented in Sec. 3, this section provides an overview of the procedural steps of our method. Initially, Algorithm 1 outlines the pretraining process for the world model. Subsequently, Algorithm 2 describes the pipeline for training and testing S-ICQL.

---

**Algorithm 1:** Pretraining the World Model

---

**Input:** Training tasks $M^{\text{train}}$ and corresponding offline datasets $\mathcal{D}^{\text{train}}$; Context encoder $E_\phi$; dynamics decoder decoder $D_\varphi$; Experience step $k$;

**for** *each iteration* **do**

    Sample a task $M^i \sim M^{\text{train}}$ and obtain the corresponding dataset $\mathcal{D}^i$ from $\mathcal{D}^{\text{train}}$

    Sample a transition tuple $(s_t, a_t, r_t, s_{t+1})$ with randomly selected $t$

    Obtain its $h$-step history $\eta_t^i = (s_{t-k}, a_{t-k}, r_{t-k}, ..., s_{t-1}, a_{t-1}, r_{t-1}, s_t, a_t)$

    Compute the context $z_t^i = E_\phi(\eta_t^i)$

    Compute the predicted reward and next state $[\hat{r}_t, \hat{s}_{t+1}] = D_\varphi(s_t, a_t; z_t^i)$ Update $E_\phi$ and $D_\varphi$

    using the loss as $\mathcal{L}(\phi, \varphi) = \mathbb{E}_{\eta_t^i \sim M^i}\left[\|[r_t, s_{t+1}] - D_\varphi(s_t, a_t; z_t^i)\|_2^2 \mid z_t^i = E_\phi(\eta_t^i)\right]$

---

## C  THE DETAILS OF ENVIRONMENTS

**DarkRoom:** The agent is randomly placed in a $10 \times 10$ grid room, and the goal occupies one of the 100 grid cells. Thus, there are 100 possible goals. The agent's observation is its current grid cell, i.e., $\mathcal{S} = [10] \times [10]$. At each step, the agent selects one of five actions: move up, down, left, right, or remain stationary. The agent receives a reward of 1 only upon reaching the goal and 0 otherwise. The episode horizon for Dark Room is 100. Consistent with (Lee et al., 2023), we use 80 of the 100 goals for pretraining and reserve the remaining 20 goals to test our model's in-context RL capability on unseen tasks.

**MuJoCo:** The multi-task MuJoCo control testbed is a classical benchmark commonly used in continual RL, multi-task RL, and meta-RL. This testbed concludes two environments with reward function changes and one environment with transition dynamics changes as

- *Cheetah-Vel*: A planar cheetah must run forward at a specified target velocity along the positive x-axis. Each task is defined by a distinct reward function that penalizes the absolute deviation between the cheetah's instantaneous velocity and its goal velocity. Goal velocities are drawn uniformly from $[0.1, 3.0]$, yielding a suite of tasks with varying targets.

- *Walker-Param*: A planar walker robot needs to move forward as fast as possible. Tasks differ in transition dynamics. For each task, the physical parameters of body mass, inertia, damping, and friction are randomized. The reward function is proportional to the running velocity in the positive

---

**Algorithm 2:** Scalable In-Context Q-Learning

---

**Input:** Training tasks $M^{\text{train}}$ and corresponding offline datasets $\mathcal{D}^{\text{train}}$; Trained context encoder $E_\phi$; Transformer model parameterized by $\theta$; Prompt horizon $h$;

\\ `Pretraining model`

**for** *each iteration* **do**

    Sample a task $M^i \sim M^{\text{train}}$ and obtain the corresponding dataset $\mathcal{D}^i$ from $\mathcal{D}^{\text{train}}$

    Sample $\left(s_t^i, a_t^i, s_{t+1}^i, r_t^i\right)$ and a $h$-step trajectory $[s_1, a_1, r_1, ..., s_h, a_h, r_h]^i$ randomly from dataset from $\mathcal{D}^i$.

    Use the trained context encoder $E_\phi$ to transform the $h$-step trajectory into lightweight prompt $\beta^i := [z_1, ..., z_h]^i = E_\phi\left([s_1, a_1, r_1, ..., s_h, a_h, r_h]^i\right)$

    \\ `In-context values learning`

    Update $V_\theta \leftarrow V_\theta - \rho\nabla_{V_\theta} L_V(\theta)$ using Eq. 5

    Update $Q_\theta \leftarrow Q_\theta - \rho\nabla_{Q_\theta} L_Q(\theta)$ using Eq. 4

    Update $\hat{\theta} \leftarrow (1-\alpha)\hat{\theta} + \alpha\theta$

    \\ `In-context policy extraction`

    Update $\pi_\theta \leftarrow \pi_\theta - \rho\nabla_{\pi_\theta} L_\pi(\theta)$ using Eq. 6

\\ `Online test-time deployment`

Sample unknown task $M^s \sim M^{\text{test}}$ and initialize empty $\beta = \{\}$.

**for** *each episode in* `max_eps` **do**

    Deploy $\pi_\theta$ by choosing $a_t \sim \pi_\theta(\cdot \mid s_t, \beta^t)$ at step $t$.

    Store $(s_t, a_t, r_t, s_{t+1})$ and use trained context encoder $E_\phi$ to transform the nearest $h$-step interactions into lightweight prompt $\beta^t$.

---

x-direction, which remains consistent for different tasks. The agent must therefore adapt its policy to diverse dynamics to achieve optimal performance.

- *Ant-Dir*: A quadrupedal ant robot must move in a specified direction. Each task defines a distinct target angle, and the reward is given by the cosine similarity between the agent's velocity vector and the unit vector in the target direction. Target directions are drawn uniformly from $[0, 2\pi]$.

- *Humanoid-Dir*: A high-dimensional humanoid agent must move in a specified direction while maintaining balance and coordinated whole-body control. Each task defines a target heading angle, and the reward is computed as the cosine similarity between the agent's velocity vector and the unit vector of the target direction. Target directions are sampled uniformly from $[0, 2\pi]$.

For all MuJoCo domains, we allocate 45 tasks for training and reserve the remaining 5 for evaluation, with each episode capped at 200 timesteps.

**Meta-World:** The Meta-Learning 1 (ML1) suite in Meta-World is a minimalist, single-task benchmark for few-shot meta-reinforcement learning. For each task, a Sawyer robotic arm in MuJoCo must reach a randomly Selected target—whose coordinates are withheld from observations—forcing the agent to infer the goal by trial and error. ML1's sparse information and clear generalization challenge make it one of Meta-World's most popular robotic manipulation testbeds for evaluating rapid adaptation in meta-RL.

- *Reach*: a Sawyer robotic arm must reach a randomly assigned goal position in 3D space, with each Reach task differing in goal location and reward function. The objective is to learn an optimal policy that efficiently generates the action sequence required to reach the specified target.

- *Push*: A Sawyer arm must push a block to a randomly placed target on a tabletop, with each task varying the block's start position and corresponding reward function. Agents receive only sparse distance-based feedback and must infer the goal through interaction. This setup evaluates the agent's ability to explore and adapt its pushing strategy under sparse supervision.

- *PickPlace*: A Sawyer arm must pick up an object and place it at a target position sampled for each task. The object's start pose and goal location vary across tasks. Rewards are sparse and reflect successful grasping and placement, making this a typical multi-step manipulation benchmark.

For all Meta-World domains, we allocate 45 tasks for training and reserve the remaining 5 for evaluation, with each episode capped at 100 timesteps.

## D  THE DETAILS OF DATASET CONSTRUCTION

**Pretraining Datasets for Darkroom:** In the Dark Room environment, the horizon of each trajectory is set to 100 steps. At each step, we follow the optimal policy with probability $\epsilon$ and a random policy with probability $1 - \epsilon$. We choose $\epsilon$ so that the mean return of the pretraining datasets is below 40% of the optimal policy return, reflecting the challenging yet common scenarios.

**Pretraining Datasets for MuJoCo and Meta-World:** For each evaluation domain, we choose 45 tasks to construct the training datasets and train a single-task policy independently for each task. We use soft actor-critic (SAC) (Haarnoja et al., 2018) for the MuJoCo domains. We use Proximal Policy Optimization (PPO) (Schulman et al., 2017) for the Meta-World domain. We collect two types of offline datasets for each evaluation domain as

- *Mixed*: The dataset is constructed by mixing data from various policies, including those saved throughout the entire training process, offering a diverse range of experiences for training.

- *Medium*: The dataset is constructed using data from medium-quality policies, which, while suboptimal compared to high-quality policies, still offer valuable experiences for training.

Table 7 and Table 8 list the main hyperparameters for the SAC and PPO algorithms during offline data collection in all evaluation domains, respectively.

Table 7: Hyperparameters of SAC used to collect multi-task datasets.

| Environments | Training steps | Warmup steps | Save frequency | Learning rate | Batch size | Soft update | Discount factor | Entropy ratio |
|---|---|---|---|---|---|---|---|---|
| Cheetah-Vel | 500000 | 2000 | 10000 | 3e-4 | 256 | 0.005 | 0.99 | 0.2 |
| Walker-Param | 1000000 | 2000 | 10000 | 3e-4 | 256 | 0.005 | 0.99 | 0.2 |
| Ant-dir | 500000 | 2000 | 10000 | 3e-4 | 256 | 0.005 | 0.99 | 0.2 |
| Humanoid-Dir | 500000 | 2000 | 10000 | 1e-4 | 256 | 0.005 | 0.99 | 0.2 |

Table 8: Hyperparameters of PPO used to collect multi-task datasets.

| Environments | Total_timesteps | n_steps | Learning_rate | Batch_size | n_epochs | Discount factor |
|---|---|---|---|---|---|---|
| Reach | 400000 | 2048 | 3e-4 | 64 | 10 | 0.99 |
| Push | 1000000 | 2048 | 3e-4 | 64 | 10 | 0.99 |
| PickPlace | 1000000 | 2048 | 3e-4 | 64 | 10 | 0.99 |

## E  THE DETAILS OF BASELINES

This section presents seven representative baselines addressing the meta-task generalization problem, including one context-based offline meta-reinforcement learning (COMRL) method and six ICRL approaches. These baselines are thoughtfully selected to span the major domains of current offline meta-task research. Furthermore, since our proposed S-ICQL method belongs to the ICRL category, we incorporate more methods from this class as baselines for a comprehensive comparison. The detailed descriptions of these baselines are as follows:

- **IC-IQL** (Tarasov et al., 2025) extends standard ICRL by explicitly optimizing reinforcement learning objectives instead of relying solely on supervised losses as in AD. It augments AD with a Q-learning loss, both the policy and value are optimized by updating only their heads, with no gradient propagation to the shared transformer backbone.

- **DIT** (Dong et al., 2024) addresses the limitations of standard autoregressive imitation learning when trained on suboptimal trajectories. Instead of treating trajectory prediction purely as supervised learning, DIT emulates an actor-critic algorithm in-context by applying a weighted

maximum likelihood estimation (WMLE) loss, where the weights are directly computed from observed rewards.

- **DICP** (Son et al., 2025), combines model-based reinforcement learning with prior in-context RL approaches such as AD and IDT. DICP leverages a pre-trained Transformer not only to condition past transitions but also to simulate future trajectories and estimate long-term returns, enabling proactive decision-making without parameter updates. DICP jointly models environment dynamics and policy improvements in context, resulting in greater adaptability and sample efficiency, particularly in long-horizon tasks.

- **IDT** (Huang et al., 2024), employs a hierarchical learning framework that decomposes decision-making across temporal scales to address the high computational cost of prior in-context RL methods on long-horizon tasks. Its architecture consists of three modules: (i) Decision-Making, which predicts high-level decisions; (ii) Decision-to-Go, which decodes these decisions into low-level actions; and (iii) Decision-Review, which maps low-level actions back to high-level representations. Built upon a transformer architecture similar to that of AD, IDT addresses complex decision-making in long-horizon tasks while reducing computational costs through hierarchical modeling.

- **AD** (Laskin et al., 2023), casts in-context RL as a supervised sequence-modeling task: a causal Transformer is pretrained on across-episodic trajectories covering the entire RL learning to predict subsequent actions, thereby emulating standard RL update dynamics without explicit gradient updates. In AD, trajectories gathered across episodes are organized into fixed-length sequences of length $H$, with each trajectory tokenized as an interleaved series of states, actions, and rewards. This allows the model to learn purely from contextual information, enabling in-context adaptation of policies to new tasks.

- **DPT** (Lee et al., 2023), adopts a supervised training paradigm to enable in-context learning for RL tasks. The core idea is to train a transformer to predict the provided optimal action for a given query state purely based on context, using interaction histories from diverse tasks as in-context datasets. DPT treats each transition tuple $(s, a, s', r)$ as a single token, rather than decomposing it into separate embeddings. This allows the attention mechanism to directly model relationships between full transitions, preserving the structural and semantic integrity of the interaction data.

- **UNICORN** (Li et al., 2024b), integrates representative methods such as FOCAL (Li et al., 2021), CORRO (Yuan & Lu, 2022), and CSRO (Gao et al., 2023), this work proposes a unified information-theoretic framework that interprets these approaches as optimizing different approximation bounds of the mutual information between task variables and their latent representations. Building on the information bottleneck principle, it further derives a general and unified objective for task representation learning, facilitating the extraction of robust and transferable task representations. This method provides a unified information-theoretic perspective that summarizes and connects several mainstream context-based offline meta-RL approaches. As it integrates key ideas from representative COMRL algorithms, it can serve as a strong and generalizable baseline.

Mainstream ICRL algorithms predominantly rely on imitation learning paradigms, which tend to learn suboptimal behaviors when pretraining data is limited in size or quality. To ensure fair comparison, we standardize the dataset size and quality across methods during pretraining. In our task setting, AD struggles to extract effective policy improvement operators due to insufficient offline data. Therefore, we adopt an AD variant that incorporates a reward-based trajectory sorting mechanism from AT (Liu & Abbeel, 2023) to better distill policy improvements. Additionally, since UNICORN under the COMRL setting requires warm-up data for task representation inference, we align it with the ICRL setup by initializing task representations using randomly sampled trajectories and updating them after each episode.

## F  Implementation Details of S-ICQL

**World Model.** In this paper, we adopt streamlined architectures for each component of the world model: the context encoder, and the dynamics decoder. The context encoder first employs a fully connected multi-layer perceptron (MLP) followed by a gated recurrent unit (GRU) network, both using ReLU activations. The GRU processes the agent's $k$ step history $\eta_t^i = (s_{t-k}, a_{t-k}, r_{t-k}, ..., s_{t-1}, a_{t-1}, r_{t-1}, s_t, a_t)^i$. and produces a 128-dimensional hidden vector. This

Table 9: The network configurations used for S-ICQL.

| World Model | Value | Causal Transformer | Value |
|---|---|---|---|
| GRU hidden dim | 128 | Layers num | 4 |
| Prompt representation dim | 16 | Attention heads num | 1 |
| Decoder hidden dim | 128 | Activation function | ReLU |
| Decoder hidden layers num | 2 | | |
| Activation function | ReLU | | |

Table 10: Hyperparameters of S-ICQL on various domains.

| Hyperparameters | Darkroom | Push | Reach | Cheetah-Vel | Walker-Param | Ant-Dir | PickPlace | Humanoid-Dir |
|---|---|---|---|---|---|---|---|---|
| Training steps | 2e5 | 4e5 | 4e5 | 4e5 | 4e5 | 4e5 | 4e5 | 8e5 |
| Learning rate | 3e-4 | 1e-4 | 1e-4 | 3e-4 | 3e-4 | 3e-4 | 1e-4 | 3e-4 |
| Prompt horizon $h$ | 100 | 100 | 100 | 200 | 200 | 200 | 200 | 200 |
| Embedding dim | 32 | 128 | 128 | 128 | 128 | 128 | 128 | 128 |
| Output layers num | 1 | 2 | 2 | 2 | 2 | 2 | 2 | 2 |
| Temperature parameter $\lambda$ | 0.01 | 0.001 | 0.001 | 0.001 | 0.1 | 0.01 | 0.01 | 0.1 |
| Expectile parameter $\omega$ | 0.7 | 0.5 | 0.5 | 0.7 | 0.7 | 0.7 | 0.5 | 0.7 |
| Soft update $\alpha$ | 0.005 | 0.005 | 0.005 | 0.005 | 0.005 | 0.005 | 0.005 | 0.005 |
| Discount factor | 0.99 | 0.99 | 0.99 | 0.99 | 0.99 | 0.99 | 0.99 | 0.99 |

vector is then projected via the MLP into a 16-dimensional task representation $z_t^i$. The reward decoder is an MLP that receives the tuple $(s_t^i, a_t^i, s_{t+1}^i, z_t^i)$ and passes it through two hidden layers of size 128 to predict the scalar reward $\hat{r}_t^i$. Analogously, the state transition decoder is an MLP that takes $(s_t^i, a_t^i, z_t^i)$ as input and uses two 128-dimensional hidden layers to predict the next state $\hat{s}_{t+1}^i$.

**Causal Transformer.** We implement S-ICQL on top of the official DPT codebase released by DPT (Lee et al., 2023) (https://github.com/jon–lee/decision-pretrained-transformer). We adhere to their architectural design, and construct the embeddings for the GPT-2 backbone as follows. Specifically, given a task dataset $\mathcal{D}^i$, we sample $(s_t^i, a_t^i, s_{t+1}^i, r_t^i)$ and a $h$-step trajectory $[s_1, a_1, r_1, ..., s_h, a_h, r_h]^i$ randomly from dataset from $\mathcal{D}^i$. Then we use the pretrained context encoder $E_\phi$ to transform raw transitions in this trajectory into precise and lightweight prompt $\beta^i := [z_1, ..., z_h]^i = E_\phi([s_1, a_1, r_1, ..., s_h, a_h, r_h]^i)$. We form state vectors $\xi_{s_t}^i = (s_t^i, 0)$, next state vectors $\xi_{s_{t+1}}^i = (s_{t+1}^i, 0)$ and state-action vectors $\xi_{\{sa\}_t}^i = (s_t^i, a_t^i, 0)$ by concatenating the relevant quantities and padding with zeros so that each $\xi^i$ has dimension $d_\xi := 2d_S + d_A + 1$. The $(h+1)$-length sequence is given by $X = (\xi^i, z_1^i, \ldots, z_h^i)$. We first apply linear projection $\texttt{Linear}(\cdot)$ to each vector and outputs the sequence $Y = (\hat{y}_0, \hat{y}_1, \ldots, \hat{y}_h)$. Then the $(h+1)$-length tokens are fed into the transformer and predict output autoregressively using a causal self-attention mask. In the output layer, we employ separate linear layers to produce the actions, the state-values, and the Q-values, respectively. In summary, Table 9 shows the details of network structures.

**Algorithm Hyperparameters.** We evaluate the proposed S-ICQL algorithm on eight environments: `Darkroom`, `Push`, `Reach`, `Cheetah-Vel`, `Walker-Param`, `Ant-Dir`, `PickPlace` and `Humanoid-Dir`. For all experiments, we use the Adam optimizer with a weight decay of 1e-4, gradient-norm clipping at 10, an experience step of 4, and a batch size of 128. Table 10 summarizes the detailed hyperparameters of S-ICQL in each domain.

**Compute.** We train our models on NVIDIA RTX3090 GPUs paired with an AMD EPYC 9654 CPU and 512GB of RAM. Pretraining the world model takes approximately 1–2 hours, while pretraining the causal transformer requires about 2–22 hours, depending on the environment's complexity.

## G   Hyperparameter Analysis

The temperature parameter $\lambda$ in Eq.(6) plays a crucial role in balancing the trade-off between behavior cloning and the greedy pursuit of high Q-values. We conduct experiments to analyze the influence of $\lambda$ on S-ICQL's performance. Figure 9 and Table 11 present the ablation results across representative domains with varying values of $\lambda$. A smaller $\lambda$ will make the distribution of advantage weights $\exp(\frac{1}{\lambda})(Q - V)$ less uniform, leading to more exploitation of Q-learning. As $\lambda$ decreases from 1.0,

Table 11: Few-shot converged results of S-ICQL with varying $\lambda$ on Mixed datasets.

| $\lambda$ | Reach | Cheetah-Vel | Ant-Dir |
|---|---|---|---|
| 1.0 | 765.99± 17.93 | -99.16± 4.00 | 486.23± 30.98 |
| 0.1 | 796.42± 7.20 | -46.54± 2.37 | 749.00± 19.55 |
| 0.01 | 799.29± 2.77 | -46.56± 3.08 | **813.34**± 14.12 |
| 0.001 | **806.97**± 6.35 | **-35.48**± 1.33 | 767.55± 19.89 |

the performance of S-ICQL can be greatly improved, highlighting the essentiality of harnessing Q-learning to steer ICRL architectures toward fundamental reward maximization. Practically, $\lambda = 0.01$ or $\lambda = 0.001$ leads to a satisfactory performance. The result further confirms the effectiveness and necessity of our in-context Q-learning.

In prompt construction, the task representation $z_t^i$ is inferred from the agent's $k$-step experience $\eta_t^i$ as $z_t^i \sim E_\phi(\eta_t^i)$. The step $k$ is crucial for capturing task-relevant information. We conduct experiments to investigate the impact of $k$ on S-ICQL's performance. Figure 10 and Table 12 present the few-shot evaluation returns of S-ICQL across representative environments with varying values of $k$. The results show that S-ICQL's performance is not sensitive to

Table 12: Few-shot converged results of S-ICQL with varying $k$ on Mixed datasets.

| $K$ | Reach | Cheetah-Vel | Ant-Dir |
|---|---|---|---|
| 2 | 802.00 ± 10.80 | -37.51 ± 1.80 | 717.95 ± 37.81 |
| 4 | **806.97** ± 6.35 | **-35.48** ± 1.33 | **813.34** ± 14.12 |
| 6 | 794.19 ± 7.93 | -40.36 ± 1.36 | 648.70 ± 56.75 |

$k$, with a moderate experience step of $k = 4$ performing the best in all evaluated domains. A small $k$ may provide insufficient task-relevant information, limiting precise in-context inference. A large $k$ may introduce redundancy and noise leading to overfitting during world model pretraining. Also, a large $k$ will increase the computation load in pretraining the world model and decrease the speed in constructing the prompt for in-context task inference. In practice, a moderate $k$ achieves fast and precise in-context inference, highlighting the superiority of our prompt design via world modeling.

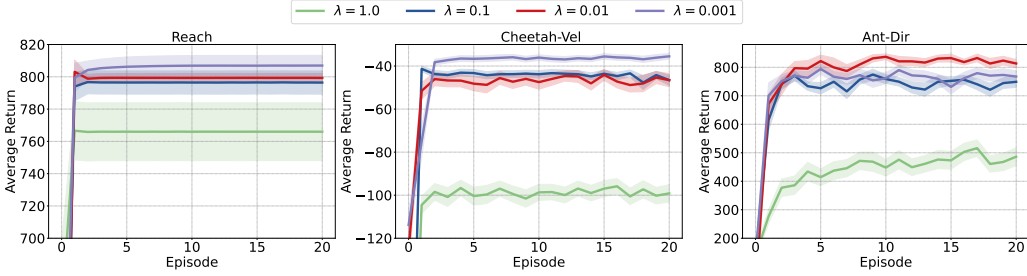

Figure 9: Few-shot evaluation return curves of S-ICQL on Mixed datasets with varying values of $\lambda$.

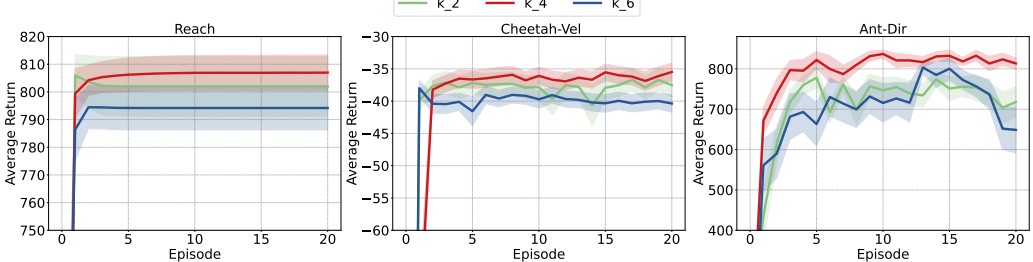

Figure 10: Few-shot evaluation return curves of S-ICQL on Mixed datasets with varying values of $k$.

## H ANALYSIS OF LOSS WEIGHT COEFFICIENTS

We conduct a hyperparameter analysis of Eq. 7 by evaluating multiple reweighting configurations for its three loss components, $\mathcal{L}_\pi(\theta)$, $\mathcal{L}_Q(\theta)$, and $\mathcal{L}_V(\theta)$. We vary the relative coefficients to emphasize different aspects of the learning objective, including configurations that up-weight the policy loss to prioritize policy fitting and configurations that increase the weight of the value or Q-value losses to strengthen critic learning, while keeping all other training settings fixed. As shown in Table 13, the performance degrades substantially when the policy loss is down-weighted (i.e., $c_1 = 0.1$), while remaining insensitive to coefficients on value losses (i.e., $c_2/c_3 = 0.1$). The equal-weight configuration $(1, 1, 1)$ consistently offers a stable and straightforward choice, supporting the use of a balanced weighting scheme for the three loss components in our main experiments.

Table 13: Analysis of loss weight coefficients in S-ICQL.

| Coefficients $(c_1, c_2, c_3)$ | $(0.1, 1, 1)$ | $(1, 0.1, 1)$ | $(1, 1, 0.1)$ | $(1, 1, 1)$ |
|---|---|---|---|---|
| Cheetah-Vel | -53.94±3.26 | -39.56±2.02 | -42.98±1.87 | **-35.48±1.33** |
| Ant-Dir | 588.81±32.00 | **843.77±8.70** | 772.04±19.11 | 813.34±14.12 |
| Reach | 591.88 ± 88.81 | 789.29 ± 9.17 | 730.37 ±38.46 | **806.97 ± 6.35** |

## I THE USE OF LARGE LANGUAGE MODELS (LLMS)

We utilize Large Language Models (LLMs) to assist with polishing the writing and improving text readability. Specifically, LLMs are employed for proofreading, enhancing grammar, and refining sentence structure. The LLM was used solely for editorial purposes to improve clarity and did not contribute to research ideation, experimental design, implementation, analysis, or scientific conclusions. All core research contributions, experiments, and analyses were conducted independently by the authors without LLM assistance.

