# OpenReview forum: "Scalable In-Context Q-Learning"
_ICLR.cc/2026/Conference — ICLR 2026 Poster_

### Official Review · Reviewer_93W6 · 2025-10-29

**Soundness:** 2
**Presentation:** 2
**Contribution:** 2
**Rating:** 4
**Confidence:** 4

**Summary:**

This work proposes scalable in-context Q-learning (S-ICQL) that adapts implicit Q-learning (IQL) to the in-context reinforcement learning (ICRL) domain in the offline setting. S-ICRL pretrains a world model to capture the dynamics of the environments, which S-ICQL leverages to encode the task context into a more compact form. The empirical study across three benchmarks confirms the effectiveness of S-ICQL.

**Strengths:**

- S-ICQL outperforms the other ICRL baselines consistently.
- The paper is overall easy to follow.
- The empirical study looks rigorous and includes the study of S-ICQL under out-of-distribution tasks and suboptimal data.

**Weaknesses:**

- I feel the comparison with other baselines is not entirely fair. For instance, algorithm distillation (AD) is more of an online method than an offline one. It requires a stream of experience generated by a consistently improving policy. In addition, the decision pre-trained transformer (DPT) requires the actions in the dataset to be optimal, which may not be satisfied in this work.
- I think there is a considerable overhead of pre-training the world model. From what I understand, S-ICQL only uses the encoder $E_\phi$ during the IQL process, and the dynamics decoder was discarded after pretraining. In addition, unlike pre-training a language model on a very general corpus, here the world model is only fitted on a particular class of tasks. Therefore, when the task class changes, the world model needs to be retrained.
- Some equations are ill-written. For example, why is there an inner expectation in equation (3)? If the encoder $E_\phi$ is a function, where does the stochasticity come from? Also, in equation (3), how is the squared norm of the tuple $[r, s]$ defined?
- In the ablation study, I don't think removing the Q-learning component recovers DPT. Without optimal actions, the model is likely regressing on the offline dataset and will naturally learn a suboptimal policy if the data quality is poor.

Minor concerns:
- Though not significantly affecting reading, the writing has several typos and grammatical errors.
- Since S-ICQL is claimed to be an ICRL algorithm, then the meta RL survey by Beck et al. (2025) and the ICRL survey by Moeini et al. (2025) should be cited for a richer context.

**Questions:**

- Why is pretraining a world model necessary for obtaining a good encoder $E_\phi$? What prevents it from learning the encoder directly during the IQL?
- If the main benefit of training the encoder is to save computation by projecting the raw transitions onto a lower-dimensional space, then why do the authors choose GRU instead of a simple linear layer?

---

> ### Author Response · Authors · 2025-11-21
> **Author Response (Part 1/2)**
>
> ***
> **`Q1. Fairness of comparison to baselines such as AD and DPT.`**
>
> A1. **We strictly ensure an entirely fair comparison with all baselines**, including AD and DPT.
>
> - **AD [1] belongs to the offline RL paradigm**, with two steps: i) collect offline datasets of learning histories from different tasks; and ii) train a causal transformer to predict actions from these histories using across-episodic contexts from offline datasets. At test time, AD algorithms are evaluated in an online mode, where online interactions with the environment act as the across-episodic contexts. In all experiments of our original paper, **we strictly maintain the same “offline training, online testing” setting for S-ICQL and all baselines**.
> - We agree that DPT [2] requires optimal dataset actions to learn an optimal policy, given its purely supervised paradigm. As noted in prior work, DPT assumes an oracle that provides optimal action labels, which is often infeasible in realistic scenarios [3]. **When only suboptimal data is available, DPT remains functional but is limited to learning suboptimal policies**. This limitation motivates us to harness dynamic programming rules to effectively learn from suboptimal data with fundamental reward maximization. In all experiments, **we test S-ICQL and DPT on the same datasets using consistent hyperparameters to ensure a strictly fair comparison**. Comprehensive results verify S-ICQL’s superiority over DPT, especially when learning from suboptimal Medium datasets.
>
> [1] In-context RL with algorithm distillation, ICLR 2023.
>
> [2] Supervised pretraining can learn in-context RL, NeurIPS 2023.
>
> [3] In-Context Reinforcement Learning From Suboptimal Historical Data, ICML 2025.
>
> ***
> **`Q2. The computation overhead of pretraining the world model.`**
>
> A2. The computational overhead of pretraining the world model is negligible compared to in-context policy training. The following table reports the time cost of the training pipeline, showing that **including the world model incurs an average of 14.60% increase in training time**. According to the ablation study in Section 4.2, **introducing the world model yields an average of 25.20% improvement in performance.** We believe that pretraining a world model is worthwhile, as it provides substantial performance gains at the cost of only a minor increase in training time.
>
> | **Time Cost** | Reach | Cheetah-Vel | Ant-Dir |
>  | :---: | :---: | :---: |  :---: |
> | **Pretraining world model** | 0.97 h | 1.95 h | 1.98 h |
> | **Training in-context policy** | 6.80 h | 13.25 h | 13.35 h |
>
> After world model pretraining, we discard the dynamics decoder since we only need the context encoder to obtain task representations as task prompts for in-context policy training. **This is a common practice in representation learning using an encoder–decoder architecture**, such as the standard autoencoder and masked autoencoder (MAE) [4]. The encoder is used to obtain compact and meaningful representations for downstream tasks, and the decoder is usually discarded.
>
> Further, we can also exploit the dynamics decoder to better leverage world model pretraining. For example, during testing in a new task, we can feed the newly collected samples into the full world model and use the dynamics prediction error to quantify the distributional discrepancy between the test task and the training distribution. We leave this as future work.
>
> [4] Masked Autoencoders Are Scalable Vision Learners, CVPR 2022.
>
> ***
> **`Q3. The world model is only fitted to a particular class of tasks.`**
>
> A3. **We strictly follow the standard ICRL setting**, or the meta-RL setting: training on a task distribution $\mathcal{T}^{\text{train}}$ and testing on a set of unseen tasks $\mathcal{T}^{\text{test}}$. Usually, the test tasks are sampled from the same distribution as the training ones, and if not, it is called out-of-distribution (OOD) testing. The world model is pretrained using the available training tasks, aiming to capture the common structure across the task distribution. Then, we can extrapolate the meta-level task representations by the extrapolation ability of the pretrained world model. **There is no need to retrain the world model for any new task within the test distribution, even for OOD tasks** (see empirical evidence in Sec. 4.3).
>
> We agree with you that a significant gap still exists between current ICRL studies and language model pretraining. The strong generalization ability of pretrained LLMs comes from scaling transformer architectures on a very large and diverse corpus. In contrast, current ICRL algorithms are trained on a relatively narrow distribution of tasks. We believe that one pressing challenge for RL practitioners is to train generalist models on a much broader range of tasks, unlocking the scaling law with the transformer architecture.

---

> ### Author Response · Authors · 2025-11-21
> **Author Response (Part 2/2)**
>
> ***
> **`Q4. The inner expectation and squared norm in Eq. (3).`**
>
> A4. **We use an inner expectation to express a more general form**. The subscript in $\mathbb{E} _ {z_t^i\sim E_{\phi}(\eta_t^i)}[\cdot]$ denotes that $z_t^i$ is sampled from the distribution derived by the encoder. Our implementation can be considered as a special case, where we use a deterministic encoder. Equivalently, $z_t^i$ is sampled from a $\delta$ distribution (the Dirac delta function, also known as the unit impulse) where the probability density is zero everywhere except at the point of the encoder’s output. In more general cases, such as variational auto-encoders, the encoder can embed the input into broader distributions like Gaussians. We agree with you that in the special case of S-ICQL’s implementation, the inner expectation can be removed.
>
> **We use square brackets $[r_t,s_t]$  to denote a concatenated vector**, other than parentheses that usually denote tuples.  The norm $||[r_t^i, s_{t+1}^i] - D_{\varphi}(s_t^i, a_t^i; z_t^i)||$ denotes the distance between the two vectors. In S-ICQL, we adopt the most commonly used Euclidean distance metric, i.e., the L2 norm. Following your advice, we have enhanced the clarity of Eq. (3) as  $\mathcal{L}(\phi, \varphi) = \mathbb{E}_ {\eta_t^i\sim M^I} \left[  \lVert [r_t,s_{t+1}] - D_ {\varphi}(s_t,a_t; z_t^i) \rVert_2^2 \mid z_t^i= E_\phi(\eta_t^i)\right]$.
>
> ***
> **`Q5. Why removing the Q-learning component reduces to DPT.`**
>
> A5. We agree with you that when removing the Q-learning component, the model degrades to the pure supervised paradigm and naturally learns a suboptimal policy from low-quality data. This exactly corresponds to the principle of DPT! As stated in our A1 to Q1, **when only suboptimal data is available, DPT remains functional but is limited to learning suboptimal policies**. This limitation serves as the key motivation of leveraging Q-learning to effectively learn from suboptimal data with fundamental reward maximization. In our experiments, all methods are trained on the same suboptimal offline datasets, simulating a realistic scenario where optimal supervision is unavailable. The superior performance of S-ICQL highlights the role of Q-learning in enabling policies to go beyond pure imitation, effectively overcoming the limitation inherent in DPT.
>
> Also, we build S-ICQL on top of DPT’s architecture, training a prompt-based transformer policy that takes the state as the query input. We implement S-ICQL on top of the official DPT codebase released at github. Strictly, S-ICQL fully reduces to DPT when both the Q-learning and world modeling components are removed.
>
> ***
> **`Q6. Minor errors and more references.`**
>
> A6. Thank you for your careful suggestions. We have thoroughly checked the minor writing errors throughout the paper, and included more related references in Related Work to enhance the literature review, such as the meta-RL survey [5] and the ICRL survey [6].
>
> [5] Beck et al. A tutorial on meta-reinforcement learning. Foundations and Trends in Machine Learning, 2025.
>
> [6] Moeini et al. A survey of in-context reinforcement learning, arXiv:2502.07978, 2025.
>
> ***
> **`Q7. Concern about the necessity of pretraining the world model for obtaining a good encoder.`**
>
> A7:  **S-ICQL separately pretrains the world model and the in-context policy to ensure better stability.** Following your suggestion, we have conducted a new ablation study that compares separate training with joint end-to-end training. **The results clearly show that world-model pretraining provides a robust and stable initial representation and learning signal for the subsequent in-context encoder**, enabling S-ICQL to achieve a 19.9% average performance improvement. Furthermore, our modular pipeline offers better scalability, since individual components can be easily substituted.
>
> | **Ablation** | **joint end to end training**  | **pretraining world model (our)**  |
> | :--- | :---: | :---: |
> | Cheetah-Vel | -51.17 $\pm$ 1.42 | **-35.48 $\pm$** 1.33 |
> | Ant-Dir | 745.88 $\pm$ 20.42 | **813.34 $\pm$** 14.12 |
>
> ***
> **`Q8. Using GRU instead of a simpler linear projection.`**
>
> A8.  **We employ a GRU specifically to capture the temporal dependencies and transition dynamics inherent in the agent's history, which a simple linear projection fails to model effectively.** Crucially, our context encoder is designed not merely for dimensionality reduction, but to abstract and encode the task-relevant information from historical experience. The intuition is that the true task belief can be inferred from the agent’s historical interactions, similar to recent meta-RL studies.  To this end, the encoder aims to infer task representation $z_t^i$ from the agent’s k-step experience $\eta_t^i=(s_{t-k},a_{t-k},r_{t-k},…,s_t,a_t)^i$ as $z_t^i=E_\phi(\eta_t^i)$. **We use a recurrent structure (e.g., GRU) that is more effective in abstracting representations from a sequence of temporal inputs than a simple linear layer.**

---

> ### Author Response · Authors · 2025-11-21
> **Author Summary Response**
>
> ***
> Thank you for your valuable review comments, which help us gain more critical insights and further enhance our empirical evaluation. We are honored to have **your positive comments on our superior performance** (“S-ICQL outperforms the other ICRL baselines consistently”), **the empirical evaluation** (“the empirical study looks rigorous, including OOD tasks and suboptimal data”), and the writing (“the paper is overall easy to follow”)
>
> Also, **other reviewers** (72Wd, gW7J, and qpLp) **provide positive evaluations of our work, including our superior performance** (“the performance of S-ICQL is strong” by 72Wd, “good experimental results which consider a number of relevant baselines” by gW7J, and “comparisons to multiple baselines demonstrating consistent improvements” by qpLp), **the empirical evaluation** (”the experiments are comprehensive” by 72Wd, “ablations show that each component contributes meaningfully…stitching section shows improvements” by gW7J, and “the experimental results are relatively comprehensive” by qpLp), **the methodological design** (“world model effectively captures task-relevant information…the integration of Q-function learning and AWR obtains better-performing behaviors” by qpLp), **and the writing** (“the manuscript is well-written and easy-to-follow” by 72Wd). We hope these collective evaluations could help clarify the breadth and strength of our contributions.
>
> We summarize that your main concerns may include the fairness of our comparisons, the training cost and generalization ability of the world model, and the clarity and justification of key design choices. **We have made a number of justifications and extended experimental analyses to address your concerns.** Also, **we have uploaded the new manuscript with modifications highlighted in blue font**. Please let us know if we have addressed your concerns. We are more than delighted to have further discussions and improve our manuscript. If our response has addressed your concerns, we would be grateful if you could re-evaluate our work.

---

### Official Review · Reviewer_qpLp · 2025-10-30

**Soundness:** 3
**Presentation:** 3
**Contribution:** 3
**Rating:** 6
**Confidence:** 4

**Summary:**

The paper proposes a new In-Context Reinforcement Learning (ICRL) framework that integrates world modeling and dynamic programming, achieving state-of-the-art results under few-shot settings on DarkRoom, MuJoCo, and Meta-World benchmarks.
The world model is used to learn task-relevant representations, which can be utilized by the learned policy.
The dynamic programming component enables the model to derive improved policies from suboptimal supervised datasets.

**Strengths:**

1.  The proposed world model effectively captures task-relevant information, providing a more compact and informative representation compared to previous ICRL approaches.

2.  The integration of  Q-function learning and advantage-weighted regression (AWR) allows the model to optimize its policy from suboptimal data and obtain better-performing behaviors.

3.  The experimental results are relatively comprehensive, with comparisons to multiple baselines demonstrating consistent improvements.

**Weaknesses:**

1.  The experimental environments are relatively  simple.
    It would strengthen the work to include results on more challenging or diverse environments.

**Questions:**

1.  After training the world model, only the encoder is used in subsequent stages.
    Are there possible ways to further utilize the world model during later training phases?

2.  The current setting trains on offline datasets and tests for generalization to OOD tasks.
    Would incorporating online learning with environment interaction in  in-distribution task improve the generalization performance on OOD tasks?

---

> ### Author Response · Authors · 2025-11-21
> **Author Response**
>
> ***
> **`Q1. Results on more challenging or diverse environments.`**
>
> A1.  Following the common practice in the ICRL community, we evaluate our method on widely adopted benchmarks, including the challenging MuJoCo and Meta-World platforms. Following your advice, we have extended our empirical evaluation to two more challenging environments: **(1) a sparse-reward and hard-exploration task (PickPlace)**, and **(2) a high-dimensional control task (Humanoid-Dir)**. These settings better reflect the complexity and diversity highlighted in your comments. As shown in the following table, **S-ICQL consistently outperforms all baseline approaches, demonstrating strong robustness and generalization in these extended environments**. Notably, the performance gap becomes even more pronounced in the high-dimensional setting, highlighting the particular strength of S-ICQL when dealing with large-scale problems, further underscoring its broader applicability. We have included full experimental details, learning curves, numerical results, and analysis in Sec. 4.7 and the updated Appendices for completeness.
>
> | **Methods** | **UNICORN** | **DPT** | **AD** | **IDT** | **DICP** | **DIT** | **IC-IQL** | **S-ICQL** |
> | :--- | :---: | :---: | :---: | :---: | :---: | :---: | :---: | :---: |
> | **PickPlace** | 0.43 $\pm$ 0.00 | 234.78 $\pm$ 20.85 | 370.09 $\pm$ 33.45 | 390.62 $\pm$42.99 | 248.54 $\pm$ 48.20 | 295.61 $\pm$ 29.79 | 386.20 $\pm$ 47.55 | **425.3 $\pm$ 61.68** |
> | **Humanoid-Dir** | 575.00 $\pm$ 22.62 | 509.99 $\pm$ 12.40 | 583.08 $\pm$ 29.64 | 615.17 $\pm$ 20.23 | 573.48 $\pm$ 21.18 | 630.06 $\pm$ 15.39 | 620.12 $\pm$ 17.17 | **728.63 $\pm$ 15.54** |
>
> ***
> **`Q2. Possible ways to further utilize the world model in later phases.`**
>
> A2.  Thank you for the insightful comment. After world model pretraining, we discard the dynamics decoder since we only need the context encoder to obtain task representations as task prompts for in-context policy training. **This is a common practice in representation learning using an encoder-decoder architecture**, such as the famous auto-encoder and masked auto-encoder (MAE) [1]. The encoders are used to obtain compact and meaningful representations for downstream tasks, and decoders are usually discarded.
>
> Further, we can also exploit the dynamics decoder to make the most use of world model pretraining. For example, **the decoder can indeed be utilized at test time to detect distributional shifts between the training tasks and newly encountered tasks.** We can feed the newly collected samples into the whole world model, and use the dynamics prediction error to quantify the distribution discrepancy between the test task and the training distribution. A significant increase in reconstruction error would indicate that the pretrained models no longer match the test task structure, suggesting the presence of a notable distribution shift. In such cases, collecting additional on-task data and updating the model becomes necessary, naturally leading to a continual or adaptive learning framework. We leave it as future work.
>
> We appreciate this suggestion and have included a dedicated discussion of this potential extension in Section 5 of the revised manuscript.
>
> [1] Masked Autoencoders Are Scalable Vision Learners, CVPR 2022.
>
> ***
> **`Q3. On the role of online interaction for improving OOD generalization.`**
>
> A3.  In the paper, we follow the common practice in the ICRL community, where the agent is trained on offline datasets collected from in-distribution tasks and evaluated on unseen or OOD test tasks. Further, incorporating online interaction within the in-distribution tasks is indeed a promising direction. In principle, additional interaction data can further enrich the task distribution and provide more diverse trajectories, which may benefit learning more generalizable in-context policies.
>
> Inspired by your insightful question, we have conducted new experiments in which the agent is first trained on the offline datasets and then allowed to collect 40 online rollouts for each in-distribution task. The online trajectories are mixed with the offline data for continued training, after which the agent is evaluated on the OOD test tasks. As shown in the following table, **this hybrid setting (offline data + limited online interaction) improves OOD generalization due to access to more informative trajectories**. However, we observe only marginal gains in OOD generalization. This suggests that although online interaction enriches the coverage of the in-distribution task manifold, it does not substantially expand the test task distribution required for stronger OOD generalization.
>
> | **Ablation** | **offline data + limited online interaction** | **offline data** |
> | :--- | :---: | :---: |
> | Cheetah-Vel | **-77.47  $\pm$** 2.44 | -83.45 $\pm$ 2.58 |
> | Ant-Dir | **674.91  $\pm$** 27.90 | 664.9  $\pm$  28.95 |

---

> ### Author Response · Authors · 2025-11-21
> **Author Summary Response**
>
> ***
> Thank you for your valuable review comments, which help us gain more critical insights and further enhance our empirical evaluation. We are honored to have **your recognition of our method** (“world model effectively captures task-relevant information…the integration of Q-function learning and AWR obtains better-performing behaviors”), **our superior performance** (“consistent improvements over multiple baselines”), **and the empirical evaluation** (“the experimental results are relatively comprehensive”).
>
> Also, **other reviewers** (72Wd, gW7J, and 93W6) **express supportive comments on several dimensions of our work, acknowledging our superior performance** (“the performance of S-ICQL is strong” by 72Wd, “good experimental results which consider a number of relevant baselines” by gW7J, and “S-ICQL outperforms the other ICRL baselines consistently” by 93W6), **the empirical evaluation** (“the experiments are comprehensive” by 72Wd, “ablations show that each component contributes meaningfully…stitching section shows improvements” by gW7J, and “the empirical study looks rigorous and includes the study under OOD tasks and suboptimal data” by 93W6), **and the writing** (“the manuscript is well-written and easy-to-follow” by 72Wd and  “the paper is overall easy to follow” by 93W6). We hope these collective evaluations could help clarify the breadth and strength of our contributions.
>
> We summarize that your main concerns may include the evaluation on more challenging environments, the potential to further utilize the world model in later phases, and the role of online interaction in improving OOD generalization. **We have made a number of justifications and experimental analyses to address your concerns.** Also, **we have uploaded the new manuscript with modifications highlighted in blue font**. Please let us know if we have addressed your concerns. We are more than delighted to have further discussions and improve our manuscript.

---

### Official Review · Reviewer_gW7J · 2025-10-31

**Soundness:** 3
**Presentation:** 2
**Contribution:** 2
**Rating:** 4
**Confidence:** 3

**Summary:**

This paper proposes S-ICQL, an in-context RL framework which encodes small chunks of experience into a compact task representation and then uses that task representation for transformer-based policy learning. Separate heads are used for the policy and value functions. Experiments show improvements over a number of ICRL/meta-RL baselines.

**Strengths:**

- Good experimental results which consider a number of relevant baselines
- Ablations show that each of the components contributes meaningfully to performance
- Stitching section shows improvements over best returns in the dataset

**Weaknesses:**

- My main issue is that it is very hard to understand the details of the method. Figure 1 is very complicated; there are many arrows going in different directions and it is not obvious what exactly “precise task representation” and “precise lightweight prompt” are.
- The method is also quite complicated and not particularly novel. It seems to be a combination of a number of existing components (e.g., AWR, world model, transformer-based policies)

**Questions:**

See weaknesses

---

> ### Author Response · Authors · 2025-11-21
> **Author Response  (Part 1/2)**
>
> ***
> **`Q1. The novelty of our method, given a a number of existing components.`**
>
> A1. **`Technical Challenges`.** Foremost, we confront two critical challenges in ICRL domains: overcoming the bottleneck of the supervised pretraining paradigm (Challenge 1) and facilitating the design of informative prompts for decision tasks (Challenge 2). Our core contribution is a scalable and parameter-efficient framework that addresses both challenges, **achieving a 14.3% to 133.7% improvement over a variety of competitive baselines**, averaged across test environments. These significant results validate the effectiveness of S-ICQL in tackling the above-mentioned challenges.
>
> **`Method contributions`.** We harness dynamic programming principles to address Challenge 1 and exploit world modeling to address Challenge 2.
>
> - **Our contribution in the first part is extending RL’s stitching property to the ICRL domain**, offering a promising avenue for realizing explicit reward maximization in in-context settings. Compared to prior works such as DIT [1], **we introduce specific designs to significantly enhance the learning process, including precise in-context inference, a unified architecture, and principled optimization**. We employ the popular IQL and AWR rules [2] to formulate a more principled framework for in-context Q-learning, efficiently mitigating value overestimation and distribution shift in offline settings. While IQL has been widely extended to broader scenarios like finetuning language models [3] and safe RL [4], its potential for in-context value function learning remains underexplored. Our work fills this gap. We do not aim to reinvent the fundamental offline RL objectives; instead, **we repurpose the well-established IQL rules to supervise efficient in-context value function learning where traditional methods may struggle.**
> - **Our contribution in the second part is to bring the idea of learning task embedding via world dynamics to the ICRL domain**. Since the world model fully characterizes the underlying task, we exploit this property to extract precise task representations as efficient task prompts. Although VariBAD [5] originally proposed learning task embeddings via world dynamics for online meta-RL, its potential for addressing precise in-context inference in ICRL remains underexplored. Similar to how subsequent studies tailored VariBAD to new challenges, such as the self-supervised learning setting [6] or the diffusion-based offline meta-RL problem [7], **we adopt the core principle of VariBAD to tackle the new ICRL challenge, enabling the design of a lightweight prompt with fast and precise in-context inference**.
>
> **`Architectural novelty`.** We design a prompt-based multi-head transformer architecture that simultaneously predicts optimal and value functions, with several appealing properties: i) **parameter efficiency,** it introduces only two lightweight heads, and the increase in parameters is negligible relative to the transformer backbone; ii) **model scalability,** a single causal transformer is trained end-to-end within the supervised pretraining paradigm, fully leveraging transformer-based scaling with a simplified pipeline; and iii) **efficient representation learning,** a shared backbone enables more coherent and informative representations for both policy and value prediction.
>
> In summary, our method is far more than a straightforward combination of existing components. **We propose a scalable and parameter-efficient ICRL framework that tackles significant challenges in the community, with substantial extensions and improvements over prior works.** We have made thorough revisions throughout the paper to clarify the technical challenges, motivations, and contributions, including concise highlights at the beginning of key paragraphs in Sec. 3.
>
> [1] In-Context Reinforcement Learning From Suboptimal Historical Data, ICML 2025.
>
> [2] Offline Reinforcement Learning with Implicit Q-Learning, ICLR 2022.
>
> [3] Offline RL for Natural Language Generation with Implicit Language Q Learning, ICLR 2023.
>
> [4] C2IQL: Constraint-Conditioned Implicit Q-learning for Safe Offline RL, ICML 2025.
>
> [5] VariBAD: A Very Good Method for Bayes-Adaptive Deep RL via Meta-Learning, ICLR 2020.
>
> [6] Meta-RL Based on Self-Supervised Task Representation Learning, AAAI 2023.
>
> [7] MetaDiffuser: Diffusion Model as Conditional Planner for Offline Meta-RL, ICML 2023.

---

> ### Author Response · Authors · 2025-11-21
> **Author Response (Part 2/2)**
>
> ***
> **`Q2. It is hard to understand the method details, with Figure 1 being complicated.`**
>
> A2. The previous version of Figure 1 attempted to include many details of the training and optimization process, such as loss backpropagation flows, advantage-weighting paths, and online interactions with the environment during testing. This fine-grained level of detail made the figure visually complex and potentially difficult to interpret. Following your advice, **we have substantially simplified the figure by only emphasizing S-ICQL’s essential components to enhance clarity**, removing fine-grained details such as backpropagation flows and re-weighting arrows.
>
> We also removed detailed technical terms, such as *precise task representation* and *lightweight prompt*, to improve the overall comprehension of Figure 1. These terms are elaborated below and are further discussed in Sec. 3.2 and Sec. 3.3.
>
> - The **precise task representation** $z$ refers to the embedding obtained by abstracting recent k-step experiences  $\eta_t^i  =  (s_{t-k}, a_{t-k}, r_{t-k},...,s_{t-1},a_{t-1},r_{t-1},s_t,a_t)^i$ into a task representation as $z^i_t = E_{\phi}(\eta_t^i)$ using the world model encoder $E_\phi$. Since the world model fully characterizes the underlying task dynamics, we exploit this property to extract **precise** information that serves as a proxy for the task identity.
> - The **lightweight prompt** refers to using a short $h$-step task representation for fast and precise in-context inference, as opposed to AD algorithms that can require long training histories as context.
>
> Thank you for your valuable concern, which greatly helps us improve the overall readability and presentation of our method. We have updated the PDF with thorough modifications accordingly, and we would appreciate it if you could let us know whether the revised version addresses your concern.
>
> ***
>
> ### `Summary Response`
>
> Thank you for your valuable review comments, which help us gain more critical insights and further enhance our empirical evaluation. We are honored to have **your positive comments on our superior performance** (“good experimental results which consider a number of relevant baselines”), **the ablation study** (“ablations show that each of the components contributes meaningfully to performance”), **and the stitching analysis** (“stitching section shows improvements over best returns in the dataset”).
>
> Also, **other reviewers** (72Wd, qpLp, and 93W6) **provide positive remarks on our overall contributions, including our superior performance** (“the performance of S-ICQL is strong” by 72Wd, “comparisons to multiple baselines demonstrating consistent improvements” by qpLp, and “S-ICQL outperforms the other ICRL baselines consistently” by 93W6), **the empirical evaluation** (“the experiments are comprehensive” by 72Wd, “the experimental results are relatively comprehensive” by qpLp, and “the empirical study looks rigorous and includes the study under OOD tasks and suboptimal data” by 93W6), **the methodological design** (“world model effectively captures task-relevant information…the integration of Q-function learning and AWR obtains better-performing behaviors” by qpLp), **and the writing** (“the manuscript is well-written and easy-to-follow” by 72Wd and “the paper is overall easy to follow” by 93W6). We hope these collective evaluations could help clarify the breadth and strength of our contributions.
>
> We summarize that your main concerns may include the clarity of our method presentation and the technical novelty of S-ICQL. **We have made specific justifications to address your concerns.** Also, **we have uploaded the new manuscript with modifications highlighted in blue font**. Please let us know if we have addressed your concerns. We are more than delighted to have further discussions and improve our manuscript. If our response has addressed your concerns, we would be grateful if you could re-evaluate our work.

---

### Official Review · Reviewer_72Wd · 2025-11-01

**Soundness:** 3
**Presentation:** 3
**Contribution:** 2
**Rating:** 4
**Confidence:** 4

**Summary:**

This work proposes Scalable In-Context Q-Learning (S-ICQL), a novel in-context reinforcement learning (ICRL) framework that harnesses dynamic programming and world modeling to steer ICRL toward efficient reward maximization and task generalization. In particular, S-ICQL incorporating RL objectives into the supervised pretraining paradigm with a transformer architecture that simultaneously predicts optimal policies and in-context value functions. In addition, S-ICQL pretrains a world model to better capture the task information for more efficient in-context inference. The authors prove the effectiveness of S-ICQL through extension experiments on standard ICRL benchmarks.

**Strengths:**

- The manuscript is well-written and easy-to-follow.
- The experiments are comprehensive, including an array of strong baselines.
- The performance of S-ICQL is strong.

**Weaknesses:**

My biggest concern is about the novelty of the proposed framework S-ICQL. While it has many components, all of its components and insights have appeared in prior works. As a consequence, S-ICQL feels like a unification of ideas from prior works. Specifically, there are two core technical contributions,
- **Section 3.3.** The world modeling is **identical** to [1]. Both the model design and pretraining process are almost identical to the task-embedding model of [1].
- **Section 3.4.** S-ICQL also proposes to estimate in-context (action) value functions and then extract the policy with advantage weighted regressions. The same idea has been implemented in [2]. The **only differences** are that (1) S-ICQL use a shared transformer backbone for the policy and value transformers, and (2) S-ICQL uses a different loss function, which is the same loss from IQL[3].

[1] Meta-DT: Offline Meta-RL as Conditional Sequence Modeling with World Model Disentanglement.

[2] In-Context Reinforcement Learning From Suboptimal Historical Data.

[3] Offline Reinforcement Learning with Implicit Q-Learning

**Questions:**

- Is it necessary to learn the in-context value functions together with the transformer policy? I think if you can first learn the in-context value functions, use the learned values functions to compute the weights for policy extraction, and lastly do the policy extraction. In this way, you don't have a moving target, i.e., the weights for AWR is fixed and this can improve training stability.
- I think it is also standard to test offline deployment performance [4], that is, performance of the performance policy conditioned on a given trajectories collected by other behavior policies.
- In Eq (7), there are no weights for the three losses? This is counter-intuitive and can be problematic if the three losses are not of the same scales.


[4] Supervised Pretraining Can Learn In-Context Reinforcement Learning

---

> ### Author Response · Authors · 2025-11-21
> **Author Response (Part 1/3)**
>
> ***
> **`Q1. The novelty of S-ICQL.`**
>
> A1. Foremost, **our core contribution is a scalable and parameter-efficient framework that tackles significant challenges in ICRL domains**: the bottleneck of the supervised paradigm and the difficulty in designing an informative prompt for decision tasks.
>
> - **Our contribution of the world modeling component is to bring the idea of learning task embedding via world dynamics to the ICRL domain**. Although VariBAD [1] originally proposed learning task embeddings via world dynamics for online meta-RL, its potential for addressing precise in-context inference in ICRL remains underexplored. Similar to how subsquent studies tailored VariBAD to new challenges, such as the diffusion-based offline meta-RL problem in MetaDiffuser [2] or the transformer-based meta-policy learning in Meta-DT [3], **we adopt the core principle of VariBAD to tackle the new ICRL challenge, enabling the design of a lightweight prompt with fast and precise in-context inference**.
> - DIT [4] is one of the most relevant baselines to our work, as it also uses a weighted maximum likelihood estimation loss to train a transformer policy. However, directly applying DIT’s standard techniques is insufficient for our setting. To bridge this gap, **S-ICQL introduces three specific designs to enhance the architecure and learning process upon DIT**: i) **Precise Inference**, compared to using raw transitions as task prompts like DIT, S-ICQL leverages world modeling to enable more precise in-context inference; ii) **Unified Architecture**, while DIT uses separate transformers to predict the policy and value functions, S-ICQL trains a single multi-head transformer policy end-to-end, releasing the transformer’s scalability and parameter efficiency with a simplified pipeline; and iii) **Principled Optimization**, while DIT uses vanilla Bellman equations to learn value functions, S-ICQL employs implicit Q-learning (IQL) rules [5] that are more principled for mitigating value overestimation and distribution shift in offline RL settings.
> - Regarding S-ICQL’s losses, we adopt the popular IQL [5] formulation due to its core superiority in handling distribution shift in offline settings. While IQL has been widely extended to broader scenarios like finetuning language models [6] and safe RL [7], its potential for in-context value function learning remains untapped. Our work fills this gap. We do not aim to reinvent the fundamental offline RL objectives; instead, **we repurpose the well-established IQL rules to supervise efficient in-context value function learning where traditional methods may struggle.**
>
> In summary, our method is far more than a straightforward unification of existing techniques. **We focus on addressing key challenges in ICRL, with significant extensions of prior works like Meta-DT and DIT.**
>
> [1] VariBAD: A Very Good Method for Bayes-Adaptive Deep RL via Meta-Learning, ICLR 2020.
>
> [2] MetaDiffuser: Diffusion Model as Conditional Planner for Offline Meta-RL, ICML 2023.
>
> [3] Meta-DT: Offline Meta-RL as Conditional Sequence Modeling via World Model Disentanglement, NeurIPS 2024.
>
> [4] In-Context Reinforcement Learning From Suboptimal Historical Data, ICML 2025.
>
> [5] Offline Reinforcement Learning with Implicit Q-Learning, ICLR 2022.
>
> [6] Offline RL for Natural Language Generation with Implicit Language Q Learning, ICLR 2023.
>
> [7] C2IQL: Constraint-Conditioned Implicit Q-learning for Safe Offline RL, ICML 2025.

---

> ### Author Response · Authors · 2025-11-21
> **Author Response (Part 2/3)**
>
> ***
> **`Q2. Concern about jointly learning the in-context policy and value functions against separate training, regarding training stability.`**
>
> A2.  Following your professional advice, we have conducted new ablation experiments comparing two settings: (1) jointly learning the in-context policy and value functions in our setting, and (2) first learning the value functions and then performing policy extraction with fixed weights.  The results in the following table show that **joint learning consistently achieves better performance**. A likely reason is that joint optimization enables the model to acquire more coherent and informative task representations, which facilitates the learning of both the policy and value functions.
>
> | **Ablation** | **S-ICQL  (joint, our setting)** | **S-ICQL (separate)** |
> | :--- | :---: | :---: |
> | Cheetah-Vel | **-35.48 $\pm$ 1.33** | -62.31 $\pm$ 4.61 |
> | Ant-Dir | **813.34  $\pm$ 14.12** | 684.81  $\pm$  19.35 |
>
> In S-ICQL, jointly learning the policy and value functions enjoys several appealing properties: i) **parameter efficiency**, it introduces only two lightweight heads compared with prior ICRL approaches such as DPT, and the parameter increase is negligible relative to the transformer backbone; ii) **model scalability**, we train a single causal transformer end-to-end within the supervised pretraining paradigm, fully releasing the transformer’s scalability with a simplified pipeline; and iii) **efficient representation learning**, a shared backbone enables more coherent and informative representation learning for both the policy and value functions. **The above empirical evidence also verifies the superiority of joint optimization in S-ICQL.**
>
> Further, in practice, we use a delayed target for value function heads to alleviate the moving target challenge and help stabilize the training process, which is a standard technique in RL algorithms like DQN and DDPG. **Comprehensive experimental results in our paper also support the feasibility of jointly training the policy and value functions with satisfactory training stability**.
>
> ***
> **`Q3. Evaluation on offline deployment settings.`**
>
> A3. Thank you for your helpful suggestion. **We primarily report online evaluation because it reflects practical deployment needs and ensures a fair comparison with other baselines, particularly the family of AD-based methods that are only evaluated in online settings.** Following your advice, we have included offline evaluations using trajectories collected by other behavior policies to provide a more comprehensive assessment. For offline testing, we randomly sampled ten trajectories for each test task from the collected data, and evaluated the performance of DPT-style methods for a fair comparison. **As illustrated in the table below, S-ICQL also demonstrates strong performance in the offline setting compared with the baselines, further validating its effectiveness across both evaluation modes**. We have updated the paper and included the offline test results in Sec. 4.1 for completeness.
>
> | Offline | Darkroom | Push  | Reach | Cheetah-Vel  | Walker-Param | Ant-Dir |
> | :--- | :---: | :---: | :---: | :---: | :---: | :---: |
> | DPT | 29.02 $\pm$   9.98 | 410.16 $\pm$ 107.03 | 783.49 $\pm$ 22.13 | -65.92$\pm$ 46.74 | 221.84 $\pm$ 103.76 | 667.34 $\pm$ 243.76 |
> | DIT | 41.82  $\pm$ 13.22 | 495.41 $\pm$ 136.7 | 805.57 $\pm$ 13.10 | -61.11 $\pm$ 35.96 | 259.07 $\pm$ 69.15  | 744.24  $\pm$144.69 |
> | S-ICQL | **76.63 $\pm$ 19.03** | **575.07  $\pm$ 118.30** | **818.04 $\pm$  8.66** | **-37.87 $\pm$ 10.76** | **414.65 $\pm$ 120.22** | **835.98  $\pm$117.25** |

---

> ### Author Response · Authors · 2025-11-21
> **Author Response (Part 3/3)**
>
> ***
> **`Q4. Analysis of loss balance coefficients in Eq. (7).`**
>
> A4.  Thank you for this thoughtful suggestion. **We stand by the equal-weight configuration $(1,1,1) $  as the most robust universal setting across tasks. Although it may not be the optimal setting for every individual task, it is by far the most stable and robust choice across all tasks**. To validate this,  we have conducted a new hyperparameter analysis to examine whether balancing coefficients are required for the three losses in Eq. (7), i.e., $\mathcal{L}(\theta) = \mathsf{c} _ 1   \mathcal{L} _ \pi(\theta) + \mathsf{c} _ 2   \mathcal{L} _ Q(\theta) + \mathsf{c} _ 3   \mathcal{L} _ V(\theta)$. We varied the coefficients to emphasize different aspects of the learning objective, including configurations that up-weight the policy loss to prioritize policy fitting and configurations that increase the weight of the value or Q-value losses to strengthen critic learning, while keeping all other training settings fixed. As shown in the table below, the performance degrades substantially when the policy loss is down-weighted (i.e., $\mathsf{c} _1=0.1$), while remaining insensitive to coefficients on value losses (i.e., $\mathsf{c} _2/\mathsf{c} _3=0.1$). **The equal-weight configuration $(1,1,1)$ consistently offers a stable and straightforward choice, consistent with our earlier findings.** We have reformulated the overall loss in a more general form in Sec. 3.5, clarified the rationale behind the equal-weight design choice, and included a new hyperparameter analysis of the loss balance coefficients in Appendix H.
>
> | coefficients $(\mathsf{c}_1,\mathsf{c}_2,\mathsf{c}_3)$ | $(0.1, 1, 1)$ | $(1, 0.1, 1)$ | $(1, 1,0.1)$ | $(1, 1, 1)$ |
> | :--- | :---: | :---: | :---: | :---: |
> | Cheetah-Vel | -53.94 $\pm$ 3.26 | -39.56 $\pm$ 2.02 | -42.98 $\pm$ 1.87 | **-35.48 $\pm$ 1.33** |
> | Ant-Dir | 588.81 $\pm$ 32.00 | **843.77 $\pm$ 8.70** | 772.04 $\pm$ 19.11 | 813.34 $\pm$ 14.12 |
> | Reach | 591.88 $\pm$ 88.81 | 789.29  $\pm$ 9.17 | 730.37 $\pm$38.46 | **806.97 $\pm$ 6.35** |
>
> ***
> ### `Summary Response`
>
> Thank you for your valuable review comments, which help us gain more critical insights and further enhance our empirical evaluation. We are honored to have **your positive comments on the empirical evaluation** (“the experiments are comprehensive, including an array of strong baselines”), **the performance of our method** (“the performance of S-ICQL is strong”), **and the writing** (“the manuscript is well-written…easy to follow”).
>
> Also, **other reviewers** (gW7J, qpLp, and 93W6) **provide additional positive feedback on our overall contributions, highlighting our superior performance** (“good experimental results…consider a number of relevant baselines” by gW7J, “comparisons to multiple baselines demonstrating consistent improvements” by qpLp, and “S-ICQL outperforms the other ICRL baselines consistently” by 93W6), **the empirical evaluation** (“ablations show that each component contributes meaningfully…stitching section shows improvements” by gW7J, “the experimental results are relatively comprehensive” by qpLp, and “the empirical study looks rigorous and includes the study under OOD tasks and suboptimal data” by 93W6), **the methodological design** (“world model effectively captures task-relevant information…the integration of Q-function learning and AWR obtains better-performing behaviors” by qpLp), **and the writing** (“the paper is overall easy to follow” by 93W6). We hope these collective evaluations could help clarify the breadth and strength of our contributions.
>
> We summarize that your main concerns may include the technical novelty of S-ICQL, the joint learning of policy and value functions, the evaluation under offline deployment with distribution shifts, and the analysis of loss balance coefficients. **We have made a number of justifications and extended experimental analyses to address your concerns.** Also, **we have uploaded the new manuscript with modifications highlighted in blue font**. Please let us know if we have addressed your concerns. We are more than delighted to have further discussions and improve our manuscript. If our response has addressed your concerns, we would be grateful if you could re-evaluate our work.

---

### Meta-Review · Area_Chair_gGr6 · 2025-12-12

**Summary:**

This work proposes a new framework to improve the performance of in-context RL agents. The framework consists of a multi-head transformer for predicting policies and value functions and additionally a world model. This is an empirical work and extensive empirical study is provided to validate the proposed approach. Extensive baselines are compared with. Careful ablation study confirms the observed remarkable performance improvements are indeed from the proposed components. The experiments on OOD tasks suggest that the proposed approach can really fulfill ICRL's promise for generalization. Overall, I think this is a solid work now and therefore recommend accept.

**Reviewer Concerns:**

There is no reviewer response to the rebuttal but I went through all reviews and rebuttals. I didn't spot any major flaws in reviews and all the raised concerns are addressed by the rebuttal in my opinion.

**Reviewer Scores:**

72Wd may increase to 6. gW7J may increase to 6. qpLp may remain 6. 93W6 may increase to 6.

---

### Decision · Program_Chairs · 2026-01-26

Accept (Poster)